# On Learning Markov Chains

**Yi HAO**
Dept. of Electrical and Computer Engineering
University of California, San Diego
La Jolla, CA 92093
yih179@ucsd.edu

**Alon Orlitsky**
Dept. of Electrical and Computer Engineering
University of California, San Diego
La Jolla, CA 92093
alon@ucsd.edu

**Venkatadheeraj Pichapati**
Dept. of Electrical and Computer Engineering
University of California, San Diego
La Jolla, CA 92093
dheerajpv7@ucsd.edu

## Abstract

The problem of estimating an unknown discrete distribution from its samples is a fundamental tenet of statistical learning. Over the past decade, it attracted significant research effort and has been solved for a variety of divergence measures. Surprisingly, an equally important problem, estimating an unknown Markov chain from its samples, is still far from understood. We consider two problems related to the min-max risk (expected loss) of estimating an unknown $k$-state Markov chain from its $n$ sequential samples: *predicting* the conditional distribution of the next sample with respect to the KL-divergence, and *estimating* the transition matrix with respect to a natural loss induced by KL or a more general $f$-divergence measure.

For the first measure, we determine the min-max prediction risk to within a linear factor in the alphabet size, showing it is $\Omega(k \log \log n / n)$ and $\mathcal{O}(k^2 \log \log n / n)$. For the second, if the transition probabilities can be arbitrarily small, then only trivial uniform risk upper bounds can be derived. We therefore consider transition probabilities that are bounded away from zero, and resolve the problem for essentially all sufficiently smooth $f$-divergences, including KL-, $L_2$-, Chi-squared, Hellinger, and Alpha-divergences.

## 1 Introduction

Many natural phenomena are inherently probabilistic. With past observations at hand, probabilistic models can therefore help us predict, estimate, and understand, future outcomes and trends. The two most fundamental probabilistic models for sequential data are *i.i.d.* processes and Markov chains. In an *i.i.d.* process, for each $i \geq 1$, a sample $X_i$ is generated independently according to the same underlying distribution. In Markov chains, for each $i \geq 2$, the distribution of sample $X_i$ is determined by just the value of $X_{i-1}$.

Let us confine our discussion to random processes over finite alphabets, without loss of generality, assumed to be $[k] := \{1, 2, \ldots, k\}$. An *i.i.d.* process is defined by a single distribution $p$ over $[k]$, while a Markov chain is characterized by a transition probability matrix $M$ over $[k] \times [k]$. We denote the initial and stationary distributions of a Markov model by $\mu$ and $\pi$, respectively. For notational consistency let $P = (p)$ denote an *i.i.d.* model and $P = (M)$ denote a Markov model.

Having observed a sample sequence $X^n := X_1, \ldots, X_n$ from an *unknown i.i.d.* process or Markov chain, a natural problem is to *predict* the next sample point $X_{n+1}$. Since $X_{n+1}$ is a random

variable, this task is typically interpreted as estimating the conditional probability distribution $P_{x^n} := \Pr(X_{n+1} = \cdot | X^n = x^n)$ of the next sample point $X_{n+1}$.

Let $[k]^*$ denote the collection of all finite-length sequences over $[k]$.

Therefore, conditioning on $X^n = x^n$, our first objective is to estimate the conditional distribution To be more precise, we would like to find an *estimator* $\hat{P}$, that associates with every sequence $x^n \in [k]^*$ a distribution $\hat{P}_{x^n}$ over $[k]$ that approximates $P_{x^n}$ in a suitable sense.

Perhaps a more classical problem is *parameter estimation*, which describes the underlying process. An *i.i.d.* process is completely characterized by $P_{x^n} = p$, hence this problem coincides with the previous one. For Markov chains, we seek to estimate the transition matrix $M$. Therefore, instead of producing a probability distribution $\hat{P}_{x^n}$, the estimator $\hat{M}$ maps every sequence $x^n \in [k]^*$ to a transition matrix $\hat{M}_{x^n}$ over $[k] \times [k]$.

For two distributions $p$ and $q$ over $[k]$, let $L(p, q)$ be the *loss* when $p$ is approximated by $q$. For the prediction problem, we measure the performance of an estimator $\hat{P}$ in terms of its *prediction risk*,

$$\rho_n^L(P, \hat{P}) := \mathop{\mathbb{E}}_{X^n \sim P}[L(P_{X^n}, \hat{P}_{X^n})] = \sum_{x^n \in [k]^n} P(x^n) L(P_{x^n}, \hat{P}_{x^n}),$$

the expected loss with respect to the sample sequence $X^n$, where $P(x^n) := \Pr(X^n = x^n)$.

For the estimation problem, we quantify the performance of the estimator by *estimation risk*. We first consider the expected loss of $\hat{M}$ with respect to a single state $i \in [k]$:

$$\mathop{\mathbb{E}}_{X^n \sim (M)}[L(M(i, \cdot), \hat{M}_{X^n}(i, \cdot))].$$

We then define the estimation risk of $\hat{M}$ given sample sequence $X^n$ as the maximum expected loss over all states,

$$\varepsilon_n^L(M, \hat{M}) := \max_{i \in [k]} \mathop{\mathbb{E}}_{X^n \sim (M)}[L(M(i, \cdot), \hat{M}_{X^n}(i, \cdot))].$$

While the process $P$ we are trying to learn is unknown, it often belongs to a known collection $\mathscr{P}$. The worst prediction risk of an estimator $\hat{P}$ over all distributions in $\mathscr{P}$ is

$$\rho_n^L(\mathscr{P}, \hat{P}) := \max_{P \in \mathscr{P}} \rho_n^L(P, \hat{P}).$$

The minimal possible worst-case prediction risk, or simply the *minimax prediction risk*, incurred by any estimator is

$$\rho_n^L(\mathscr{P}) := \min_{\hat{P}} \rho_n^L(\mathscr{P}, \hat{P}) = \min_{\hat{P}} \max_{P \in \mathscr{P}} \rho_n^L(P, \hat{P}).$$

The *worst-case estimation risk* $\varepsilon_n^L(\mathscr{P}, \hat{M})$ and the *minimax estimation risk* $\varepsilon_n^L(\mathscr{P})$ are defined similarly. Given $\mathscr{P}$, our goals are to approximate the minimax prediction/estimation risk to a universal constant-factor, and to devise estimators that achieve this performance.

An *alternative* definition of the estimation risk, considered in [1] and mentioned by a reviewer, is

$$\tilde{\varepsilon}_n^L(M, \hat{M}) := \sum_{i \in [k]} \pi_i \cdot \mathop{\mathbb{E}}_{X^n \sim (M)}[L(M(i, \cdot), \hat{M}_{X^n}(i, \cdot))].$$

We denote the corresponding *minimax estimation risk* by $\tilde{\varepsilon}_n^L(\mathscr{P})$.

*Let $o(1)$ represent a quantity that vanishes as $n \to \infty$. In the following, we use $a \lesssim b$ to denote $a \leq b(1 + o(1))$, and $a \asymp b$ to denote $a \leq b(1 + o(1))$ and $b \leq a(1 + o(1))$.*

For the collection $\mathbb{IID}^k$ of all the *i.i.d.* processes over $[k]$, the above two formulations coincide and the problem is essentially the classical discrete distribution estimation problem. The problem of determining $\rho_n^L(\mathbb{IID}^k)$ was introduced by [2] and studied in a sequence of papers [3, 4, 5, 6, 7]. For fixed $k$ and KL-divergence loss, as $n$ goes to infinity, [7] showed that

$$\rho_n^{\text{KL}}(\mathbb{IID}^k) \asymp \frac{k - 1}{2n}.$$

KL-divergence and many other important similarity measures between two distributions can be expressed as $f$-divergences [8]. Let $f$ be a convex function with $f(1) = 0$, the $f$-divergence between two distributions $p$ and $q$ over $[k]$, whenever well-defined, is $D_f(p,q) := \sum_{i \in [k]} q(i) f(p(i)/q(i))$. Call an $f$-divergence *ordinary* if $f$ is thrice continuously differentiable over $(0, \infty)$, sub-exponential, namely, $\lim_{x \to \infty} |f(x)|/e^{cx} = 0$ for all $c > 0$, and satisfies $f''(1) \neq 0$.

Observe that all the following notable measures are ordinary $f$-divergences: Chi-squared divergence [9] from $f(x) = (x-1)^2$, KL-divergence [10] from $f(x) = x \log x$, Hellinger divergence [11] from $f(x) = (\sqrt{x} - 1)^2$, and Alpha-divergence [12] from $f_\alpha(x) := 4(1 - x^{(1+\alpha)/2})/(1 - \alpha^2)$, where $\alpha \neq \pm 1$.

**Related Work**  For any $f$-divergence, we denote the corresponding minimax prediction risk for an $n$-element sample over set $\mathscr{P}$ by $\rho_n^f(\mathscr{P})$. Researchers in [13] considered the problem of determining $\rho_n^f(\mathbb{IID}^k)$ for the ordinary $f$-divergences. Except the above minimax formulation, recently, researchers also considered formulations that are more adaptive to the underlying *i.i.d.* processes [14] [15]. Surprisingly, while the min-max risk of *i.i.d.* processes was addressed in a large body of work, the risk of Markov chains, which frequently arise in practice, was not studied until very recently.

Let $\mathbb{M}^k$ denote the collection of all the Markov chains over $[k]$. For prediction with KL-divergence, [16] showed that $\rho_n^{\text{KL}}(\mathbb{M}^k) = \Theta_k (\log \log n/n)$, but did not specify the dependence on $k$. For estimation, [17] considered the class of Markov Chains whose pseudo-spectral gap is bounded away from 0 and approximated the $L_1$ estimation risk to within a $\log n$ factor. Some of their techniques, in particular the lower-bound construction in their displayed equation $(4.3)$, are of similar nature and were derived independently of results in Section 5 in our paper.

Our first main result determines the dependence of $\rho_n^{\text{KL}}(\mathbb{M}^k)$ on both $k$ and $n$, to within a factor of roughly $k$:

**Theorem 1**  *The minimax KL-prediction risk of Markov chains satisfies*

$$\frac{(k-1) \log \log n}{4en} \lesssim \rho_n^{KL}(\mathbb{M}^k) \lesssim \frac{2k^2 \log \log n}{n}.$$

Depending on $M$, some states may be observed very infrequently, or not at all. This does not drastically affect the prediction problem as these states will be also have small impact on $\rho_n^{\text{KL}}(\mathbb{M}^k)$ in the prediction risk $\rho_n^L(P, \hat{P})$. For estimation, however, rare and unobserved states still need to be well approximated, hence $\varepsilon_n^L(\mathbb{M}^k)$ does not decrease with $n$, and for example $\varepsilon_n^{\text{KL}}(\mathbb{M}^k) = \log k$ for all $n$.

We therefore parametrize the risk by the lowest probability in the transition matrix. For $\delta > 0$ let

$$\mathbb{M}_\delta^k := \{(M) : M_{i,j} \geq \delta, \ \forall i, j\},$$

be the collection of Markov chains whose lowest transition probability exceeds $\delta$. Note that $\mathbb{M}_\delta^k$ is trivial if $\delta \geq 1/k$, we only consider $\delta \in (0, 1/k)$. We characterize the minimax estimation risk of $\mathbb{M}_\delta^k$ almost precisely.

**Theorem 2**  *For all ordinary $f$-divergences and all $\delta \in (0, 1/k)$,*

$$\tilde{\varepsilon}_n^f(\mathbb{M}_\delta^k) \asymp \frac{(k-1)k f''(1)}{2n}$$

*and*

$$(1 - \delta) \frac{(k-2) f''(1)}{2n\delta} \lesssim \varepsilon_n^f(\mathbb{M}_\delta^k) \lesssim \frac{(k-1) f''(1)}{2n\delta}.$$

We can further refine the estimation-risk bounds by partitioning $\mathbb{M}_\delta^k$ based on the smallest probability in the chain's stationary distribution $\pi$. Clearly, $\min_{i \in [k]} \pi_i \leq 1/k$. For $0 < \pi^* \leq 1/k$ and $0 < \delta < 1/k$, let

$$\mathbb{M}_{\delta, \pi^*}^k := \{(M) : (M) \in \mathbb{M}_\delta^k \text{ and } \min_{i \in [k]} \pi_i = \pi^*\}$$

be the collection of all Markov chains in $\mathbb{M}_\delta^k$ whose lowest stationary probability is $\pi^*$. We determine the minimax estimation risk over $\mathbb{M}_{\delta, \pi^*}^k$ nearly precisely.

**Theorem 3** *For all ordinary f-divergences,*

$$(1 - \pi^*)\frac{(k-2)kf''(1)}{2n} \lesssim \tilde{\varepsilon}_n^f(\mathbb{M}_{\delta,\pi^*}^k) \lesssim \frac{(k-1)kf''(1)}{2n}$$

*and*

$$(1 - \pi^*)\frac{(k-2)f''(1)}{2n\pi^*} \lesssim \varepsilon_n^f(\mathbb{M}_{\delta,\pi^*}^k) \lesssim \frac{(k-1)f''(1)}{2n\pi^*}.$$

For $L_2$-distance corresponding to the squared Euclidean norm, we prove the following risk bounds.

**Theorem 4** *For all $\delta \in (0, 1/k)$,*

$$\tilde{\varepsilon}_n^{L_2}(\mathbb{M}_\delta^k) \asymp \frac{k-1}{n}$$

*and*

$$(1 - \delta)^2\frac{1 - \frac{1}{k-1}}{n\delta} \lesssim \varepsilon_n^{L_2}(\mathbb{M}_\delta^k) \lesssim \frac{1 - \frac{1}{k}}{n\delta}.$$

**Theorem 5** *For all $\delta \in (0, 1/k)$ and $\pi^* \in (0, 1/k]$,*

$$(1 - \pi^*)^2\frac{k - \frac{k}{k-1}}{n} \lesssim \tilde{\varepsilon}_n^{L_2}(\mathbb{M}_{\delta,\pi^*}^k) \lesssim \frac{k-1}{n}$$

*and*

$$(1 - \pi^*)^2\frac{1 - \frac{1}{k-1}}{n\pi^*} \lesssim \varepsilon_n^{L_2}(\mathbb{M}_{\delta,\pi^*}^k) \lesssim \frac{1 - \frac{1}{k}}{n\pi^*}.$$

The rest of the paper is organized as follows. Section 2 introduces add-constant estimators and additional definitions and notation for Markov chains. Note that each of the above results consists of a lower bound and an upper bound. We prove the lower bound by constructing a suitable prior distribution over the relevant collection of processes. Section 3 and 5 describe these prior distributions for the prediction and estimation problems, respectively. The upper bounds are derived via simple variants of the standard add-constant estimators. Section 4 and 6 describe the estimators for the prediction and estimation bounds, respectively. For space considerations, we relegate all the proofs to the supplemental material.

## 2 Definitions and Notation

### 2.1 Add-constant estimators

Given a sample sequence $X^n$ from an *i.i.d.* process $(p)$, let $N_i'$ denote the number of times symbol $i$ appears in $X^n$. The classical *empirical estimator* estimates $p$ by

$$\hat{p}_{X^n}(i) := \frac{N_i'}{n}, \; \forall i \in [k].$$

The empirical estimator performs poorly for loss measures such as KL-divergence. For example, if $p$ assigns a tiny probability to a symbol so that it is unlikely to appear in $X^n$, then with high probability the KL-divergence between $p$ and $\hat{p}_{X^n}$ will be infinity.

A common solution applies the Laplace smoothing technique [18] that assigns to each symbol $i$ a probability proportional to $N_i' + \beta$, where $\beta > 0$ is a fixed constant. The resulting *add-$\beta$* estimator, is denoted by $\hat{p}^{+\beta}$. Due to their simplicity and effectiveness, add-$\beta$ estimators are widely used in various machine learning algorithms such as naive Bayes classifiers [19]. As shown in [7], for the *i.i.d.* processes, a variant of the add-3/4 estimator achieves the minimax estimation risk $\rho_n^{\text{KL}}(\mathbb{IID}^k)$.

Analogously, given a sample sequence $X^n$ generated by a Markov chain, let $N_{ij}$ denote the number of times symbol $j$ appears right after symbol $i$ in $X^n$, and let $N_i$ denote the number of times that symbol $i$ appears in $X^{n-1}$. We define the add-$\beta$ estimator $\hat{M}^{+\beta}$ as

$$\hat{M}_{X^n}^{+\beta}(i, j) := \frac{N_{ij} + \beta}{N_i + k\beta}, \; \forall i, j \in [k].$$

## 2.2 More on Markov chains

Adopting notation in [20], let $\Delta_k$ denote the collection of discrete distributions over $[k]$. Let $[k]^e$ and $[k]^o$ be the collection of even and odd integers in $[k]$, respectively. By convention, for a Markov chain over $[k]$, we call each symbol $i \in [k]$ a *state*. Given a Markov chain, the *hitting time* $\tau(j)$ is the first time the chain reaches state $j$. We denote by $\Pr_i(\tau(j) = t)$ the probability that starting from $i$, the hitting time of $j$ is exactly $t$. For a Markov chain $(M)$, we denote by $P^t$ the distribution of $X_t$ if we draw $X^t \sim (M)$. Additionally, for a fixed Markov chain $(M)$, the *mixing time* $t_{mix}$ denotes the smallest index $t$ such that $L_1(P^t, \pi) < 1/2$. Finally, for notational convenience, we write $M_{ij}$ instead of $M(i, j)$ whenever appropriate.

## 3 Minimax prediction: lower bound

A standard lower-bound argument for minimax prediction risk uses the fact that

$$\rho_n^{\mathrm{KL}}(\mathscr{P}) = \min_{\hat{P}} \max_{P \in \mathscr{P}} \rho_n^{\mathrm{KL}}(P, \hat{P}) \geq \min_{\hat{P}} \mathbb{E}_{P \sim \Pi}[\rho_n^{\mathrm{KL}}(P, \hat{P})]$$

for any prior distribution $\Pi$ over $\mathscr{P}$. One advantage of this approach is that the optimal estimator that minimizes $\mathbb{E}_{P \sim \Pi}[\rho_n^{\mathrm{KL}}(P, \hat{P})]$ can often be computed explicitly.

Perhaps the simplest prior is the uniform distribution $U(\mathscr{P}_S)$ over a subset $\mathscr{P}_S \subset \mathscr{P}$. Let $\hat{P}^*$ be the optimal estimator minimizing $\mathbb{E}_{P \sim U(\mathscr{P}_S)}[\rho_n^{\mathrm{KL}}(P, \hat{P})]$. Computing $\hat{P}^*$ for all the possible sample sequences $x^n$ may be unrealistic. Instead, let $\mathscr{K}_n$ be an arbitrary subset of $[k]^n$, we can lower bound

$$\rho_n^{\mathrm{KL}}(P, \hat{P}) = \mathbb{E}_{X^n \sim P}[D_{\mathrm{KL}}(P_{X^n}, \hat{P}_{X^n})]$$

by

$$\rho_n^{\mathrm{KL}}(P, \hat{P}; \mathscr{K}_n) := \mathbb{E}_{X^n \sim P}[D_{\mathrm{KL}}(P_{X^n}, \hat{P}_{X^n}) \mathbb{1}_{X^n \in \mathscr{K}_n}].$$

Hence,

$$\rho_n^{\mathrm{KL}}(\mathscr{P}) \geq \min_{\hat{P}} \mathbb{E}_{P \sim U(\mathscr{P}_S)}[\rho_n^{\mathrm{KL}}(P, \hat{P}; \mathscr{K}_n)].$$

The key to applying the above arguments is to find a proper pair $(\mathscr{P}_S, \mathscr{K}_n)$.

Without loss of generality, assume that $k$ is even. Let $a := \frac{1}{n}$ and $b := 1 - \frac{k-2}{n}$, and define

$$M_n(p_2, p_4, \dots, p_k) := \begin{bmatrix} b-a & a & a & a & \dots & a & a \\ p_2 & b{-}p_2 & a & a & \dots & a & a \\ a & a & b-a & a & \dots & a & a \\ a & a & p_4 & b{-}p_4 & \dots & a & a \\ \vdots & \vdots & \vdots & \vdots & \ddots & \vdots & \vdots \\ a & a & a & a & \dots & b-a & a \\ a & a & a & a & \dots & p_k & b{-}p_k \end{bmatrix}.$$

In addition, let

$$V_n := \left\{ \frac{1}{\log^t n} : t \in \mathbb{N} \text{ and } 1 \leq t \leq \frac{\log n}{2 \log \log n} \right\},$$

and let $u_k$ denote the uniform distribution over $[k]$. Finally, given $n$, define

$$\mathscr{P}_S = \{(M) \in \mathbb{M}^k : \mu = u_k \text{ and } M = M_n(p_2, p_4, \dots, p_k), \text{ where } p_i \in V_n, \forall i \in [k]^e\}.$$

Next, let $\mathscr{K}_n$ be the collection of sequences $x^n \in [k]^n$ whose last appearing state didn't transition to any other state. For example, 3132, or 31322, but not 21323. In other words, for any state $i \in [k]$, let $\bar{i}$ represent an arbitrary state in $[k] \setminus \{i\}$, then

$$\mathscr{K}_n = \{x^n \in [k]^n : x^n = \bar{i}^{n-\ell} i^{\ell} : i \in [k], n-1 \geq \ell \geq 1\}.$$

## 4    Minimax prediction: upper bound

For the $\mathscr{K}_n$ defined in the last section,

$$\rho_n^{\mathrm{KL}}(P, \hat{P}; \mathscr{K}_n) = \sum_{x^n \in \mathscr{K}_n} P(x^n) D_{\mathrm{KL}}(P_{x^n}, \hat{P}_{x^n}).$$

We denote the *partial minimax prediction risk* over $\mathscr{K}_n$ by

$$\rho_n^{\mathrm{KL}}(\mathscr{P}; \mathscr{K}_n) := \min_{\hat{P}} \max_{P \in \mathscr{P}} \rho_n^{\mathrm{KL}}(P, \hat{P}; \mathscr{K}_n).$$

Let $\overline{\mathscr{K}_n} := [k]^n \setminus \mathscr{K}_n$. Define $\rho_n^{\mathrm{KL}}(P, \hat{P}; \overline{\mathscr{K}_n})$ and $\rho_n^{\mathrm{KL}}(\mathscr{P}; \overline{\mathscr{K}_n})$ in the same manner. As the consequence of $\hat{P}$ being a function from $[k]^n$ to $\Delta_k$, we have the following triangle inequality,

$$\rho_n^{\mathrm{KL}}(\mathscr{P}) \le \rho_n^{\mathrm{KL}}(\mathscr{P}; \overline{\mathscr{K}_n}) + \rho_n^{\mathrm{KL}}(\mathscr{P}; \mathscr{K}_n).$$

Turning back to Markov chains, let $\hat{P}^{+\frac{1}{2}}$ denote the estimator that maps $X^n \sim (M)$ to $\hat{M}^{+\frac{1}{2}}(X_n, \cdot)$, one can show that

$$\rho_n^{\mathrm{KL}}(\mathbb{M}^k; \overline{\mathscr{K}_n}) \le \max_{P \in \mathbb{M}^k} \rho_n^{\mathrm{KL}}(P, \hat{P}^{+\frac{1}{2}}; \overline{\mathscr{K}_n}) \le \mathcal{O}_k\left(\tfrac{1}{n}\right).$$

Recall the following lower bound

$$\rho_n^{\mathrm{KL}}(\mathbb{M}^k) = \Omega_k\left(\frac{\log \log n}{n}\right).$$

This together with the above upper bound on $\rho_n^{\mathrm{KL}}(\mathbb{M}^k; \overline{\mathscr{K}_n})$ and the triangle inequality shows that an upper bound on $\rho_n^{\mathrm{KL}}(\mathbb{M}^k; \mathscr{K}_n)$ also suffices to bound the leading term of $\rho_n^{\mathrm{KL}}(\mathbb{M}^k)$. The following construction yields such an upper bound.

We partition $\mathscr{K}_n$ according to the last appearing state and the number of times it transitions to itself,

$$\mathscr{K}_n = \cup_{\ell=1}^{n-1} K_\ell(i), \text{ where } K_\ell(i) := \{x^n \in [k]^n : x^n = \bar{i}^{n-\ell} i^\ell\}.$$

For any $x^n \in \mathscr{K}_n$, there is a unique $K_\ell(i)$ such that $x^n \in K_\ell(i)$. Consider the following estimator

$$\hat{P}_{x^n}(i) := \begin{cases} 1 - \frac{1}{\ell \log n} & \ell \le \frac{n}{2} \\ 1 - \frac{1}{\ell} & \ell > \frac{n}{2} \end{cases}$$

and

$$\hat{P}_{x^n}(j) := \frac{1 - \hat{P}_{x^n}(i)}{k-1}, \ \forall j \in [k] \setminus \{i\},$$

we can show that

$$\rho_n^{\mathrm{KL}}(\mathbb{M}^k; \mathscr{K}_n) \le \max_{P \in \mathbb{M}^k} \rho_n^{\mathrm{KL}}(P, \hat{P}; \mathscr{K}_n) \lesssim \frac{2k^2 \log \log n}{n}.$$

The upper-bound proof applies the following lemma that uniformly bounds the hitting probability of any $k$-state Markov chain.

**Lemma 1**    *[21] For any Markov chain over $[k]$ and any two states $i, j \in [k]$, if $n > k$, then*

$$Pr_i(\tau(j) = n) \le \frac{k}{n}.$$

## 5    Minimax estimation: lower bound

Analogous to Section 3, we use the following standard argument to lower bound the minimax risk

$$\varepsilon_n^L(\mathscr{M}) = \min_{\hat{M}} \max_{(M) \in \mathscr{M}} \varepsilon_n^L(M, \hat{M}) \ge \min_{\hat{M}} \mathop{\mathbb{E}}_{(M) \sim U(\mathscr{M}_S)} [\varepsilon_n^L(M, \hat{M})],$$

where $\mathscr{M}_S \subset \mathscr{M}$ and $U(\mathscr{M}_S)$ is the uniform distribution over $\mathscr{M}_S$. Setting $\mathscr{M} = \mathbb{M}^k(\delta, \pi^*)$, we outline the construction of $\mathscr{M}_S$ as follows.

Let $u_{k-1}$ be the uniform distribution over $[k-1]$. As in [13], denote the $L_\infty$ ball of radius $r$ around $u_{k-1}$ by

$$B_{k-1}(r) := \{p \in \Delta_{k-1} : L_\infty(p, u_{k-1}) < r\},$$

where $L_\infty(\cdot, \cdot)$ is the $L_\infty$ distance between two distributions. Define

$$p' := (p_1,\ p_2,\ \ldots,\ p_{k-1}),$$

$$p^* := \left( \frac{\bar{\pi}^*}{k-1},\ \frac{\bar{\pi}^*}{k-1},\ \cdots\ \frac{\bar{\pi}^*}{k-1},\ \pi^* \right),$$

and

$$M_n(p') := \begin{bmatrix} \frac{\bar{\pi}^*}{k-1} & \frac{\bar{\pi}^*}{k-1} & \cdots & \frac{\bar{\pi}^*}{k-1} & \pi^* \\ \frac{\pi^*}{k-1} & \frac{\pi^*}{k-1} & \cdots & \frac{\pi^*}{k-1} & \pi^* \\ \vdots & \vdots & \ddots & \vdots & \vdots \\ \frac{\pi^*}{k-1} & \frac{\pi^*}{k-1} & \cdots & \frac{\pi^*}{k-1} & \pi^* \\ \bar{\pi}^* p_1 & \bar{\pi}^* p_2 & \cdots & \bar{\pi}^* p_{k-1} & \pi^* \end{bmatrix},$$

where $\bar{\pi}^* = 1 - \pi^*$ and $\sum_{i=1}^{k-1} p_i = 1$.

Given $n$ and $\epsilon \in (0, 1)$, let $n' := (n(1+\epsilon)\pi^*)^{1/5}$. We set

$$\mathscr{M}_S = \{(M) \in \mathbb{M}^k(\delta, \pi^*) : \mu = p^* \text{ and } M = M_n(p'), \text{ where } p' \in B_{k-1}(1/n')\}.$$

Noting that the uniform distribution over $\mathscr{M}_S$, $U(\mathscr{M}_S)$, is induced by $U(B_{k-1}(1/n'))$, the uniform distribution over $B_{k-1}(1/n')$ and thus is well-defined.

An important property of the above construction is that for a sample sequence $X^n \sim (M) \in \mathscr{M}_S$, $N_k$, the number of times that state $k$ appears in $X^n$, is a binomial random variable with parameters $n$ and $\pi^*$. Therefore, by the following lemma, $N_k$ is highly concentrated around its mean $n\pi^*$.

**Lemma 2** *[22] Let $Y$ be a binomial random variable with parameters $m \in \mathbb{N}$ and $p \in [0, 1]$, then for any $\epsilon \in (0, 1)$,*

$$Pr(Y \geq (1+\epsilon)mp) \leq \exp\left(-\epsilon^2 mp/3\right).$$

In order to prove the lower bound on $\tilde{\varepsilon}_n^f(\mathbb{M}_{\delta,\pi^*}^k)$, we only need to modify the above construction as follows. Instead of drawing the last row of the transition matrix $M_n(p')$ uniformly from the distribution induced by $U(B_{k-1}(1/n'))$, we draw all rows independently in the same fashion. The proof is omitted due to similarity.

## 6   Minimax estimation: upper bound

The proof of the upper bound relies on a concentration inequality for Markov chains in $\mathbb{M}_\delta^k$, which can be informally expressed as

$$\Pr(|N_i - (n-1)\pi(i)| > t) \leq \Theta_\delta(\exp(\Theta_\delta(-t^2/n))).$$

Note that this inequality is very similar to the Hoeffding's inequality for *i.i.d.* processes.

The difficulty in analyzing the performance of the original add-$\beta$ estimator is that the chain's initial distribution could be far away from its stationary distribution and finding a simple expression for $\mathbb{E}[N_i]$ and $\mathbb{E}[N_{ij}]$ could be hard. To overcome this difficulty, we ignore the first few sample points and construct a new add-$\beta$ estimator based on the remaining sample points. Specifically, let $X^n$ be a length-$n$ sample sequence drawn from the Markov chain $(M)$. Removing the first $m$ sample points, $X_{m+1}^n := X_{m+1}, \ldots, X_n$ can be viewed as a length-$(n-m)$ sample sequence drawn from $(M)$ whose initial distribution $\mu'$ satisfies

$$L_1(\mu', \pi) < 2(1-\delta)^{m-1}.$$

Let $m = \sqrt{n}$. For sufficiently large $n$, $L_1(\mu', \pi) \ll 1/n^2$ and $\sqrt{n} \ll n$. Hence without loss of generality, we assume that the original initial distribution $\mu$ already satisfies $L_1(\mu, \pi) < 1/n^2$. If not, we can simply replace $X^n$ by $X_{\sqrt{n}+1}^n$.

To prove the desired upper bound for ordinary $f$-divergences, it suffices to use the add-$\beta$ estimator

$$\hat{M}_{X^n}^{+\beta}(i,j) := \frac{N_{ij} + \beta}{N_i + k\beta}, \ \forall i, j \in [k].$$

For the $L_2$-distance, instead of an add-constant estimator, we apply an add-$\sqrt{N_i}/k$ estimator

$$\hat{M}_{X^n}^{+\sqrt{N_i}/k}(i,j) := \frac{N_{ij} + \sqrt{N_i}/k}{N_i + \sqrt{N_i}}, \ \forall i, j \in [k].$$

## 7 Experiments

We augment the theory with experiments that demonstrate the efficacy of our proposed estimators and validate the functional form of the derived bounds.

We briefly describe the experimental setup. For the first three figures, $k = 6$, $\delta = 0.05$, and $10,000 \leq n \leq 100,000$. For the last figure, $\delta = 0.01$, $n = 100,000$, and $4 \leq k \leq 36$. In all the experiments, the initial distribution $\mu$ of the Markov chain is drawn from the $k$-Dirichlet(1) distribution. For the transition matrix $M$, we first construct a transition matrix $M'$ where each row is drawn independently from the $k$-Dirichlet(1) distribution. To ensure that each element of $M$ is at least $\delta$, let $J_k$ represent the $k \times k$ all-ones matrix, and set $M = M'(1 - k\delta) + \delta J_k$. We generate a new Markov chain for each curve in the plots. And each data point on the curve shows the average loss of 100 independent restarts of the same Markov chain.

The plots use the following abbreviations: Theo for theoretical minimax-risk values; Real for real experimental results: using the estimators described in Sections 4 and 6; Pre for average prediction loss and Est for average estimation loss; Const for add-constant estimator; Prop for proposed add-$\sqrt{N_i}/k$ estimator described in Section 6; Hell, Chi, and Alpha(c) for Hellinger divergence, Chi-squared divergence, and Alpha-divergence with parameter $c$. In all three graphs, the theoretical min-max curves are precisely the upper bounds in the corresponding theorems, except that in the prediction curve in Figure 1a the constant factor 2 in the upper bound is adjusted to $1/2$ to better fit the experiments. Note the excellent fit between the theoretical bounds and experimental results.

Figure 1a shows the decay of the experimental and theoretical KL-prediction and KL-estimation losses with the sample size $n$. Figure 1b compares the $L_2$-estimation losses of our proposed estimator and the add-one estimator, and the theoretical minimax values. Figure 1c compares the experimental estimation losses and the theoretical minimax-risk values for different loss measures. Finally, figure 1d presents an experiment on KL-learning losses that scales $k$ up while $n$ is fixed. All the four plots demonstrate that our theoretical results are accurate and can be used to estimate the loss incurred in learning Markov chains. Additionally, Figure 1b shows that our proposed add-$\sqrt{N_i}/k$ estimator is uniformly better than the traditional add-one estimator for different values of sample size $n$. We have also considered add-constant estimators with different constants varying from 2 to 10 and our proposed estimator outperformed all of them.

## 8 Conclusions

We studied the problem of learning an unknown $k$-state Markov chain from its $n$ sequential sample points. We considered two formulations: prediction and estimation. For prediction, we determined the minimax risk up to a multiplicative factor of $k$. For estimation, when the transition probabilities are bounded away from zero, we obtained nearly matching lower and upper bounds on the minimax risk for $L_2$ and ordinary $f$-divergences. The effectiveness of our proposed estimators was verified through experimental simulations. Future directions include closing the gap in the prediction problem in Section 1, extending the results on the min-max estimation problem to other classes of Markov chains, and extending the work from the classical setting $k \ll n$, to general $k$ and $n$.

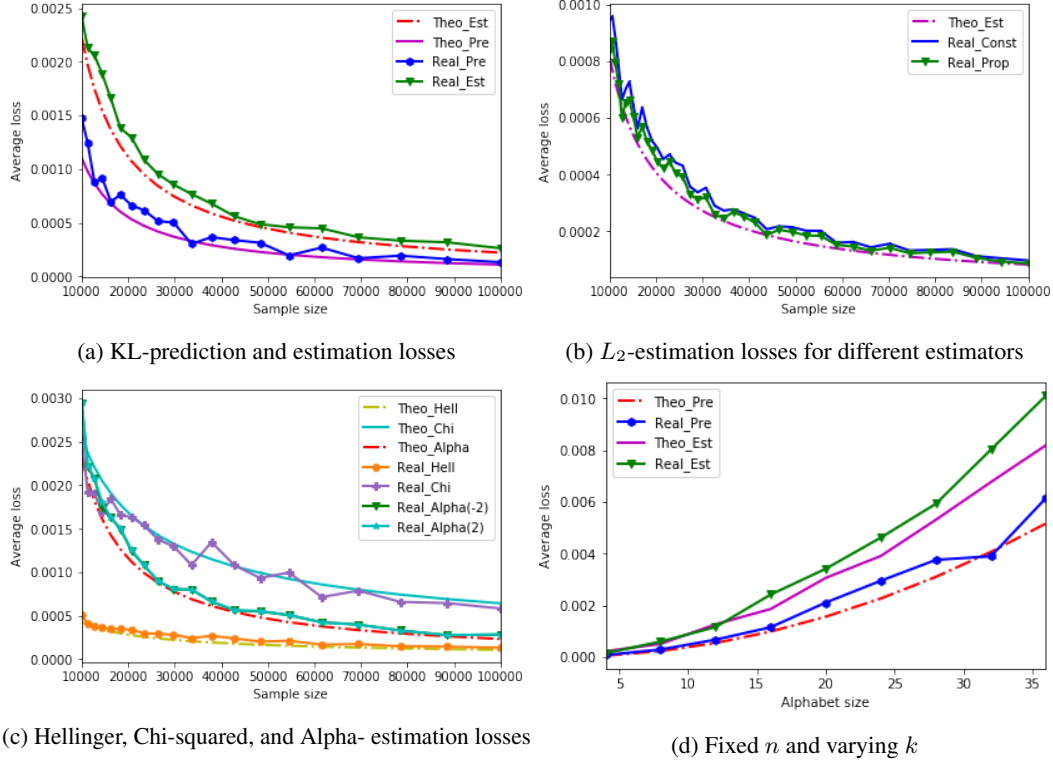

(a) KL-prediction and estimation losses

(b) $L_2$-estimation losses for different estimators

(c) Hellinger, Chi-squared, and Alpha- estimation losses

(d) Fixed $n$ and varying $k$

Figure 1: Experiments

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
