[Supplementary Material]

# Supplemental: Minimax Learning of Markov Chains

**Yi HAO**
University of California, San Diego
yih179@eng.ucsd.edu

**Alon Orlitsky**
University of California, San Diego
alon@eng.ucsd.edu

**Venkatadheeraj Pichapati**
University of California, San Diego
dheerajpv7@eng.ucsd.edu

**Additional notation**   For simplicity, we shall write $c = a \pm b$ instead of $c \in [a - b, a + b]$ whenever appropriate. We use $a \gtrsim b$ and $a \lesssim b$ to denote $a \geq b(1 + o(1))$ and $a \leq b(1 + o(1))$, respectively.

## 1   Minimax prediction: lower bound

A standard argument for lower bounding the minimax prediction risk is

$$\rho_n^{\text{KL}}(\mathscr{P}) = \min_{\hat{P}} \max_{P \in \mathscr{P}} \rho_n^{\text{KL}}(P, \hat{P}) \geq \min_{\hat{P}} \mathbb{E}_{P \sim \Pi}[\rho_n^{\text{KL}}(P, \hat{P})],$$

where $\Pi$ is a prior distribution over $\mathscr{P}$. The advantage of this approach is that the optimal estimator that minimizes $\mathbb{E}_{P \sim \Pi}[\rho_n^{\text{KL}}(P, \hat{P})]$ can often be computed explicitly.

Perhaps the simplest prior is the uniform distribution over some subset of $\mathscr{P}$. Consider the uniform distribution over $\mathscr{P}_S \subset \mathscr{P}$, say $U(\mathscr{P}_S)$, the following lemma shows an explicit way of computing the optimal estimator for $\mathbb{E}_{P \sim U(\mathscr{P}_S)}[\rho_n^{\text{KL}}(P, \hat{P})]$ when $\mathscr{P}_S$ is finite.

**Lemma 1** *Let $\hat{P}^*$ be the optimal estimator that minimizes $\mathbb{E}_{P \sim U(\mathscr{P}_S)}[\rho_n^{KL}(P, \hat{P})]$, then for any $x^n \in [k]^n$ and any symbol $i \in [k]$,*

$$\hat{P}_{x^n}^*(i) = \sum_{P \in \mathscr{P}_S} \frac{P(x^n)}{\sum_{P' \in \mathscr{P}_S} P'(x^n)} P_{x^n}(i).$$

Clearly, computing $\hat{P}^*$ for all the possible sample sequences $x^n$ may be unrealistic. Instead, let $\mathscr{K}_n$ be an arbitrary subset of $[k]^n$, we can lower bound

$$\rho_n^{\text{KL}}(P, \hat{P}) = \mathbb{E}_{X^n \sim P}[D_{\text{KL}}(P_{X^n}, \hat{P}_{X^n})]$$

by

$$\rho_n^{\text{KL}}(P, \hat{P}; \mathscr{K}_n) := \mathbb{E}_{X^n \sim P}[D_{\text{KL}}(P_{X^n}, \hat{P}_{X^n}) \mathbb{1}_{X^n \in \mathscr{K}_n}].$$

This yields

$$\rho_n^{\text{KL}}(\mathscr{P}) \geq \min_{\hat{P}} \mathbb{E}_{P \sim U(\mathscr{P}_S)}[\rho_n^{\text{KL}}(P, \hat{P}; \mathscr{K}_n)].$$

The key to apply the above arguments is to find a proper pair $(\mathscr{P}_S, \mathscr{K}_n)$. The rest of this section is organized as follows. In Subsection 1.1, we present our construction of $\mathscr{P}_S$ and $\mathscr{K}_n$. In Subsection 1.2, we find the exact form of the optimal estimator using Lemma 1. Then we analyze its prediction risk over $\mathscr{K}_n$ in Subsection 1.3, where we further partition $\mathscr{K}_n$ into smaller subsets $K_\ell(i)$, and lower bound the KL-divergence over $K_\ell(i)$ and the probability $P(X^n \in K_\ell(i))$ in Lemma 4 and 5, respectively. Finally, we consolidate all the previous results and prove the desired lower bound on $\rho_n^{\text{KL}}(\mathscr{P})$.

## 1.1 Prior construction

Without loss of generality, we assume that $k$ is an even integer. For notational convenience, we denote by $u_k$ the uniform distribution over $[k]$ and define

$$M_n(p_2, p_4, \ldots, p_k) := \begin{bmatrix} b-a & a & a & a & \ldots & a & a \\ p_2 & b-p_2 & a & a & \ldots & a & a \\ a & a & b-a & a & \ldots & a & a \\ a & a & p_4 & b-p_4 & \ldots & a & a \\ \vdots & \vdots & \vdots & \vdots & \ddots & \vdots & \vdots \\ a & a & a & a & \ldots & b-a & a \\ a & a & a & a & \ldots & p_k & b-p_k \end{bmatrix},$$

where $a := \frac{1}{n}$ and $b := 1 - \frac{k-2}{n}$. In addition, let

$$V_n := \left\{ \frac{1}{\log^t n} : t \in \mathbb{N} \text{ and } 1 \le t \le \frac{\log n}{2 \log \log n} \right\}.$$

Given $n$, we set

$$\mathscr{P}_S = \{(M) \in \mathbb{M}^k : \mu = u_k \text{ and } M = M_n(p_2, p_4, \ldots, p_k), \text{ where } p_i \in V_n, \forall i \in [k]^e\}.$$

Then, we choose $\mathscr{K}_n$ to be the collection of sequences $x^n \in [k]^n$ whose last appearing state didn't transition to any other symbol. In other words, for any state $i \in [k]$, let $\bar{i}$ represent an arbitrary state other than $i$, then

$$\mathscr{K}_n = \{x^n \in [k]^n : x^n = \bar{i}^{n-\ell} i^\ell : i \in [k], n-1 \ge \ell \ge 1\}.$$

According to both the last appearing state and the number of times it transitions to itself, we can partition $\mathscr{K}_n$ as

$$\mathscr{K}_n = \cup_{\ell=1}^{n-1} K_\ell(i), \text{ where } K_\ell(i) := \{x^n \in [k]^n : x^n = \bar{i}^{n-\ell} i^\ell\}.$$

## 1.2 The optimal estimator

Let $\hat{P}^*$ denote the optimal estimator that minimizes $\mathbb{E}_{P \sim U(\mathscr{P}_S)}[\rho_n^{\mathrm{KL}}(P, \hat{P}; \mathscr{K}_n)]$. The following lemma presents the exact form of $\hat{P}^*$.

**Lemma 2** *For any $x^n \in \mathscr{K}_n$, there exists a unique $K_\ell(i)$ that contains it. Consider $\hat{P}_{x^n}^*$, we have:*

1. *If $i \in [k]^e$, then*

$$\hat{P}_{x^n}^*(j) := \begin{cases} a & j > i \text{ or } j < i-1 \\ \sum_{v \in V_n}(b-v)^\ell / \sum_{v \in V_n}(b-v)^{\ell-1} & j = i \\ \sum_{v \in V_n}(b-v)^{\ell-1}v / \sum_{v \in V_n}(b-v)^{\ell-1} & j = i-1 \end{cases}$$

2. *If $i \in [k]^o$, then*

$$\hat{P}_{x^n}^*(j) := \begin{cases} a & j > i \text{ or } j < i \\ b-a & j = i \end{cases}$$

**Proof** Given $(M) \in \mathscr{P}_S$, consider $X^n \sim (M)$,

$$\Pr(X^n = x^n) = \frac{1}{k} \prod_{i_1 \in [k]} \prod_{j_1 \in [k]} M_{i_1 j_1}^{N_{i_1 j_1}}.$$

By Lemma 1, for any $x^n \in K_\ell(i)$ and $j \in [k]$, $\hat{P}_{x^n}^*(j)$ evaluates to

$$\hat{P}_{x^n}^*(j) = \frac{\displaystyle\sum_{(M) \in \mathscr{P}_S} M_{ij} \prod_{i_1 \in [k]} \prod_{j_1 \in [k]} M_{i_1 j_1}^{N_{i_1 j_1}}}{\displaystyle\sum_{(M) \in \mathscr{P}_S} \prod_{i_1 \in [k]} \prod_{j_1 \in [k]} M_{i_1 j_1}^{N_{i_1 j_1}}}.$$

Noting that $x^n \in K_\ell(i)$ implies $N_{ii} = \ell - 1$ and $N_{ij} = 0, \forall j \neq i$. Besides, for any $j_1 \in [k]$ and $i_1 \in [k] \setminus \{j_1, j_1 + 1\}$, $M_{i_1 j_1}$ is uniquely determined by $i_1$ and $j_1$ for all $(M) \in \mathscr{P}_S$.

Thus, for $s = 0$ or $1$, we can rewrite $M_{ij}{}^s \prod_{i_1 \in [k]} \prod_{j_1 \in [k]} M_{i_1 j_1}^{N_{i_1 j_1}}$ as

$$C(x^n, k) M_{ij}^s \prod_{\substack{t=2 \\ t \text{ even}}}^{k} \left[M_{t(t-1)}\right]^{N_{t(t-1)}} \left[M_{tt}\right]^{N_{tt}},$$

where $C(x^n, k)$ is a constant that only depends on $x^n$ and $k$.

Hence, for any $x^n \in K_\ell(i)$,

$$\hat{P}_{x^n}^*(j) = \frac{\displaystyle\sum_{(M) \in \mathscr{P}_S} M_{ij} \prod_{\substack{t=2 \\ t \text{ even}}}^{k} \left[M_{t(t-1)}\right]^{N_{t(t-1)}} \left[M_{tt}\right]^{N_{tt}}}{\displaystyle\sum_{(M) \in \mathscr{P}_S} \prod_{\substack{t=2 \\ t \text{ even}}}^{k} \left[M_{t(t-1)}\right]^{N_{t(t-1)}} \left[M_{tt}\right]^{N_{tt}}}.$$

Below we show how to evaluate $\hat{P}_{x^n}^*(j)$ for $j = i \in [k]^e$, and other cases can be derived similarly.

Combining $M_{jj}{}^{N_{jj}}$ with $M_{jj}$ in the nominator,

$$
\begin{aligned}
\hat{P}_{x^n}^*(j) &= \frac{\displaystyle\sum_{(M) \in \mathscr{P}_S} \left[M_{jj}^\ell\right] \prod_{\substack{t=2 \\ t \text{ even} \\ t \neq j}}^{k} \left[M_{t(t-1)}\right]^{N_{t(t-1)}} \left[M_{tt}\right]^{N_{tt}}}{\displaystyle\sum_{(M) \in \mathscr{P}_S} \left[M_{jj}^{\ell-1}\right] \prod_{\substack{t=2 \\ t \text{ even} \\ t \neq j}}^{k} \left[M_{t(t-1)}\right]^{N_{t(t-1)}} \left[M_{tt}\right]^{N_{tt}}} \\[2ex]
&= \frac{\displaystyle\sum_{\substack{v \in V_n \\ v' \in V_n}} (b - v')^\ell \prod_{\substack{t=2 \\ t \text{ even} \\ t \neq j}}^{k} v^{N_{t(t-1)}} (b-v)^{N_{tt}}}{\displaystyle\sum_{\substack{v \in V_n \\ v' \in V_n}} (b - v')^{\ell-1} \prod_{\substack{t=2 \\ t \text{ even} \\ t \neq j}}^{k} v^{N_{t(t-1)}} (b-v)^{N_{tt}}} \\[2ex]
&= \frac{\left[\sum_{v' \in V_n} (b-v')^\ell\right] \displaystyle\sum_{v \in V_n} \prod_{\substack{t=2 \\ t \text{ even} \\ t \neq j}}^{k} v^{N_{t(t-1)}} (b-v)^{N_{tt}}}{\left[\sum_{v' \in V_n} (b-v')^{\ell-1}\right] \displaystyle\sum_{v \in V_n} \prod_{\substack{t=2 \\ t \text{ even} \\ t \neq j}}^{k} v^{N_{t(t-1)}} (b-v)^{N_{tt}}} \\[2ex]
&= \frac{\sum_{v \in V_n} (b-v)^\ell}{\sum_{v \in V_n} (b-v)^{\ell-1}}.
\end{aligned}
$$

This completes the proof.

## 1.3 Analysis

Next, for any $x^n \in K_\ell(i)$, we lower bound $D_{\text{KL}}(P_{x^n}, \hat{P}_{x^n}^*)$ in terms of $M_{i(i-1)}$ and $\hat{P}_{x^n}^*(i-1)$.

**Lemma 3** *For any* $(M) \in \mathscr{P}_S$ *and* $x^n \in K_\ell(i)$,

$$D_{KL}(P_{x^n}, \hat{P}^*_{x^n}) \geq M_{i(i-1)}\left(-1 + \log \frac{M_{i(i-1)}}{\hat{P}^*_{x^n}(i-1)}\right).$$

**Proof**    By the previous lemma,

$$D_{\mathrm{KL}}(P_{x^n}, \hat{P}^*_{x^n}) = M_{ii} \log \frac{M_{ii}}{\hat{P}^*_{x^n}(i)} + M_{i(i-1)} \log \frac{M_{i(i-1)}}{\hat{P}^*_{x^n}(i-1)}.$$

Noting that $\frac{x}{x+1} \leq \log(x+1)$ for all $x > -1$,

$$\begin{aligned}
M_{ii} \log \frac{M_{ii}}{\hat{P}^*_{x^n}(i)} &= M_{ii} \log\left(\frac{M_{ii} - \hat{P}^*_{x^n}(i)}{\hat{P}^*_{x^n}(i)} + 1\right) \\
&\geq M_{ii} - \hat{P}^*_{x^n}(i) \\
&= \left(b - M_{i(i-1)}\right) - \left(b - \hat{P}^*_{x^n}(i-1)\right) \\
&\geq -M_{i(i-1)}.
\end{aligned}$$

This completes the proof.

Let $V'_n := \{\frac{1}{(\log n)^t} \mid t \in \mathbb{N}, 1 \leq t \leq \frac{\log n}{4 \log \log n}\}$ be a subset of $V_n$ whose size is $\frac{1}{2}|V_n|$. For $M_{i(i-1)} \in V'_n$, we further lower bound $M_{i(i-1)}/\hat{P}^*_{x^n}(i-1)$ in terms of $n$.

Let $\ell_1(M) := \frac{1}{M_{i(i-1)}} \frac{1}{\log \log n}$ and $\ell_2(M) := \frac{1}{M_{i(i-1)}} \log \log n$, we have

**Lemma 4** *For any* $(M) \in \mathscr{P}_S$, $x^n \in K_\ell(i)$ *where* $i \in [k]^e$, $M_{i(i-1)} = \frac{1}{(\log n)^m} \in V'_n$, *and sufficiently large* $n$, *if*

$$\ell_1(M) \leq \ell \leq \ell_2(M),$$

*then,*

$$\frac{M_{i(i-1)}}{\hat{P}^*_{x^n}(i-1)} \gtrsim \frac{\log n}{8 \log \log n}(1 - o(1)).$$

**Proof**    Consider $M_{i(i-1)} = \frac{1}{(\log n)^m} \in V'_n$, where $m \in [1, \frac{\log n}{4 \log \log n}]$.

Note that for $x^n \in K_\ell(i)$, the value of $\hat{P}^*_{x^n}(i-1)$ only depends on $\ell$, we can define

$$F_\ell := \frac{M_{i(i-1)}}{\hat{P}^*_{x^n}(i-1)}.$$

We have

$$F_\ell \geq \frac{A_\ell + X_\ell + C_\ell}{B_\ell + X_\ell + D_\ell},$$

where

$$X_\ell := \left(1 - \frac{k-2}{n} - \frac{1}{(\log n)^m}\right)^\ell,$$

$$A_\ell := \sum_{i=1}^{m-1}\left(1 - \frac{k-2}{n} - \frac{1}{(\log n)^i}\right)^\ell,$$

$$C_\ell := \sum_{i=m+1}^{\frac{\log n}{2\log\log n}}\left(1 - \frac{k-2}{n} - \frac{1}{(\log n)^i}\right)^\ell,$$

$$B_\ell := \sum_{i=1}^{m-1}\left(1 - \frac{k-2}{n} - \frac{1}{(\log n)^i}\right)^\ell (\log n)^{m-i},$$

$$\text{and } D_\ell := \sum_{i=m+1}^{\frac{\log n}{2\log\log n}}\left(1 - \frac{k-2}{n} - \frac{1}{(\log n)^i}\right)^\ell (\log n)^{m-i}.$$

We have the following bounds on these quantities.

**Bounds for $X_\ell$**

$$0 \le X_\ell = \left(1 - \frac{k-2}{n} - \frac{1}{(\log n)^m}\right)^\ell \le 1.$$

**Bounds for $A_\ell$**

$$0 \le A_\ell = \sum_{i=1}^{m-1} \left(1 - \frac{k-2}{n} - \frac{1}{(\log n)^i}\right)^\ell.$$

**Bounds for $D_\ell$**

$$0 \le D_\ell \le \sum_{i=m+1}^{\frac{\log n}{2 \log \log n}} \left(1 - \frac{k-2}{n} - \frac{1}{(\log n)^i}\right)^\ell \frac{1}{\log n} = \frac{1}{\log n} C_\ell.$$

**Bounds for $C_\ell$**

Note that

$$\frac{(\log n)^m}{\log \log n} \le \ell \le (\log n)^m \log \log n$$

and

$$(\log n)^m \le \sqrt{n}.$$

Consider a single term of $C_\ell$, we have

$$
\begin{aligned}
\left(1 - \frac{k-2}{n} - \frac{1}{(\log n)^i}\right)^\ell &\ge \left(1 - \frac{k-2}{n} - \frac{1}{(\log n)^i}\right)^{(\log n)^m \log \log n} \\
&= \left(1 - \frac{k-2}{n} - \frac{1}{(\log n)^i}\right)^{\frac{1}{\frac{k-2}{n} + \frac{1}{(\log n)^i}} \left(\frac{k-2}{n} + \frac{1}{(\log n)^i}\right)(\log n)^m \log \log n} \\
&\ge \left[\left(1 - \frac{k-2}{n} - \frac{1}{(\log n)^i}\right)^{\frac{1}{\frac{k-2}{n} + \frac{1}{(\log n)^i}}}\right]^{\left(\frac{k-2}{\sqrt{n}} + \frac{1}{\log n}\right) \log \log n} \\
&\ge \left(\frac{1}{4}\right)^{\frac{(k-2) \log \log n}{\sqrt{n}} + \frac{\log \log n}{\log n}} \\
&\ge \left(\frac{1}{4}\right)^{\frac{1}{2}} = \frac{1}{2},
\end{aligned}
$$

where we use the inequality $i \ge m+1$ and $(1 - \frac{1}{x})^x \ge \frac{1}{4}$ for $x \ge 2$.

Hence,

$$\frac{\log n}{8 \log \log n} = \frac{\log n}{4 \log \log n} \cdot \frac{1}{2} \le C_\ell = \sum_{i=m+1}^{\frac{\log n}{2 \log \log n}} \left(1 - \frac{k-2}{n} - \frac{1}{(\log n)^i}\right)^\ell \le \sum_{i=m+1}^{\frac{\log n}{2 \log \log n}} 1 \le \frac{\log n}{2 \log \log n}.$$

**Bounds for $B_\ell$**

Similarly, consider a single term of $B_\ell$ without the factor $(\log n)^{m-i}$,

$$\left(1 - \frac{k-2}{n} - \frac{1}{(\log n)^i}\right)^\ell \leq \left(1 - \frac{k-2}{n} - \frac{1}{(\log n)^i}\right)^{\frac{(\log n)^m}{\log\log n}}$$

$$\leq \left(1 - \frac{k-2}{n} - \frac{1}{(\log n)^i}\right)^{\frac{1}{\frac{1}{(\log)^i} + \frac{k-2}{n}}\left(\frac{1}{(\log)^i} + \frac{k-2}{n}\right)\frac{(\log n)^m}{\log\log n}}$$

$$\leq \left[\left(1 - \frac{k-2}{n} - \frac{1}{(\log n)^i}\right)^{\frac{1}{\frac{1}{(\log)^i} + \frac{k-2}{n}}}\right]^{\left(\frac{1}{(\log)^i} + \frac{k-2}{n}\right)\frac{(\log n)^m}{\log\log n}}$$

$$\leq \left(\frac{1}{e}\right)^{\frac{(\log n)^{m-i}}{\log\log n}}$$

$$= \left(\frac{1}{n}\right)^{\frac{(\log n)^{m-i-1}}{\log\log n}},$$

where we use the inequality $(1 - \frac{1}{x})^x \leq \frac{1}{e}$ for $x \geq 2$.

Hence,

$$B_\ell = \sum_{i=1}^{m-1}\left(1 - \frac{k-2}{n} - \frac{1}{(\log n)^i}\right)^\ell (\log n)^{m-i}$$

$$\leq \sum_{i=1}^{m-1}\left(\frac{1}{n}\right)^{\frac{(\log n)^{m-i-1}}{\log\log n}}(\log n)^{m-i}$$

$$= \sum_{i=1}^{m-1}\left(\frac{1}{n}\right)^{\frac{(\log n)^{m-i-1}}{3\log\log n}}(\log n)^{m-i}\left(\frac{1}{n}\right)^{\frac{2(\log n)^{m-i-1}}{3\log\log n}}$$

$$= \left(\frac{1}{n}\right)^{\frac{1}{\log\log n}}\log n + \sum_{i=1}^{m-2}\left(\frac{1}{n}\right)^{\frac{(\log n)^{m-i-1}}{3\log\log n}}(\log n)^{m-i}\left(\frac{1}{n}\right)^{\frac{2(\log n)^{m-i-1}}{3\log\log n}}$$

$$\leq \left(\frac{1}{n}\right)^{\frac{1}{\log\log n}}\log n + \sum_{i=1}^{m-2}\left(\frac{1}{n}\right)^{\frac{\log n}{3\log\log n}}(\log n)^m\left(\frac{1}{n}\right)^{\frac{2\log n}{3\log\log n}}$$

$$\leq \left(\frac{1}{n}\right)^{\frac{1}{\log\log n}}\log n + \sum_{i=1}^{m-2}\left(\frac{1}{n}\right)^{\frac{\log n}{3\log\log n}}(\log n)^{\frac{\log n}{4\log\log n}}\left(\frac{1}{n}\right)^{\frac{2\log n}{3\log\log n}}$$

$$\leq \left(\frac{1}{n}\right)^{\frac{1}{\log\log n}}\log n + \frac{\log n}{4\log\log n}\left(\frac{1}{n}\right)^{\frac{\log n}{3\log\log n}}(\log n)^{\frac{\log n}{4\log\log n}}\left(\frac{1}{n}\right)^{\frac{2\log n}{3\log\log n}}$$

$$\leq \left(\frac{1}{n}\right)^{\frac{1}{\log\log n}}\log n + \left(\frac{1}{n}\right)^{\frac{\log n}{3\log\log n}}(\log n)^{\frac{\log n}{4\log\log n}+1}\left(\frac{1}{n}\right)^{\frac{2\log n}{3\log\log n}}$$

$$\leq \left(\frac{1}{n}\right)^{\frac{1}{\log\log n}}e^{\log\log n} + \left(\frac{\log n}{n}\right)^{\frac{\log n}{3\log\log n}}\left(\frac{1}{n}\right)^{\frac{2\log n}{3\log\log n}}$$

$$\leq e^{\frac{-\log n + (\log\log n)^2}{\log\log n}} + \frac{1}{n}$$

$$\leq e^{-\frac{2(\log\log n)^2 + (\log\log n)^2}{\log\log n}} + \frac{1}{n}$$

$$\leq e^{-\log\log n} + \frac{1}{n}$$

$$\leq \frac{2}{\log n},$$

where we use the inequality $x - 2(\log x)^2 \geq 0$ for $x \geq 1$.

Putting everything together:

$$F_\ell = \frac{A_\ell + X_\ell + C_\ell}{B_\ell + X_\ell + D_\ell} \geq \frac{0 + \frac{\log n}{8 \log \log n}}{\frac{2}{\log n} + 1 + \frac{1}{2 \log \log n}} \asymp \frac{\log n}{8 \log \log n}.$$

This completes the proof.

Another quantity that will be appear later is $\Pr(X^n \in K_\ell(i))$ where $X^n \sim (M) \in \mathscr{P}_S$. We need the following lower bound.

**Lemma 5** *For $X^n \sim (M) \in \mathscr{P}_S$ and $i \in [k]^e$,*

$$\Pr(X^n \in K_\ell(i)) \gtrsim \frac{k-1}{ek} \frac{1}{n} \left(1 - \frac{k-2}{n} - M_{i(i-1)}\right)^{l-1}.$$

**Proof**   By our construction of $\mathscr{P}_S$, for $X^n \sim (M) \in \mathscr{P}_S$ and $i \in [k]^e$, we have the following observations.

1. The probability that the initial state is not $i$ is $\frac{k-1}{k}$.

2. The probability of transitioning from some state $j \neq i$ to some state that is not $i$ is $1 - \frac{1}{n}$.

3. The probability of transitioning from some state $j \neq i$ to state $i$ is $\frac{1}{n}$.

4. The probability of transitioning from state $i$ to itself is $1 - \frac{k-2}{n} - M_{i(i-1)}$.

Therefore,

$$\Pr(X^n \in K_\ell(i)) = \frac{k-1}{k} \left(1 - \frac{1}{n}\right)^{n-\ell-1} \frac{1}{n} \left(1 - \frac{k-2}{n} - M_{i(i-1)}\right)^{\ell-1}$$

$$\geq \frac{k-1}{k} \left(1 - \frac{1}{n}\right)^{n} \frac{1}{n} \left(1 - \frac{k-2}{n} - M_{i(i-1)}\right)^{\ell-1}$$

$$\asymp \frac{k-1}{ek} \frac{1}{n} \left(1 - \frac{k-2}{n} - M_{i(i-1)}\right)^{\ell-1}.$$

This completes the proof.

Now we turn back to $\rho_n^{\mathrm{KL}}(\mathscr{P})$. According to the previous derivations,

$$\rho_n^{\mathrm{KL}}(\mathscr{P}) \geq \min_{\hat{P}} \mathbb{E}_{P \sim U(\mathscr{P}_S)}[\rho_n^{\mathrm{KL}}(P, \hat{P}; \mathscr{K}_n)]$$

$$= \mathbb{E}_{P \sim U(\mathscr{P}_S)}\left[\sum_{x^n \in \mathscr{K}_n} \Pr_{X^n \sim P}(X^n = x^n) D_{\mathrm{KL}}(P_{x^n}, \hat{P}_{x^n}^*)\right]$$

$$= \frac{1}{|\mathscr{P}_S|} \sum_{(M) \in \mathscr{P}_S} \sum_{l=1}^{n-1} \sum_{i \in [k]} \sum_{x^n \in K_\ell(i)} \left[\Pr_{X^n \sim P}(X^n = x^n) D_{\mathrm{KL}}(P_{x^n}, \hat{P}_{x^n}^*)\right]$$

$$\geq \frac{1}{|\mathscr{P}_S|} \sum_{(M) \in \mathscr{P}_S} \sum_{\ell = \ell_1(M)}^{\ell_2(M)} \sum_{i \in [k]^e} \sum_{x^n \in K_\ell(i)} \left[\Pr_{X^n \sim P}(X^n = x^n) D_{\mathrm{KL}}(P_{x^n}, \hat{P}_{x^n}^*)\right].$$

Noting that all $x^n \in K_\ell(i)$ have the same $P_{x^n}$ and $\hat{P}_{x^n}^*$, thus, the last formula can be written as

$$\frac{1}{|\mathscr{P}_S|} \sum_{(M) \in \mathscr{P}_S} \sum_{\ell = \ell_1(M)}^{\ell_2(M)} \sum_{i \in [k]^e} \left[\Pr_{X^n \sim P}(X^n \in K_\ell(i)) D_{\mathrm{KL}}(P_{x^n}, \hat{P}_{x^n}^*; x^n \in K_\ell(i))\right].$$

By Lemma 3 and 4, for $\ell_1(M) \leq \ell \leq \ell_2(M)$ and $M_{i(i-1)} \in V_n'$,

$$D_{\mathrm{KL}}(P_{x^n}, \hat{P}_{x^n}^*; x^n \in K_\ell(i)) \geq M_{i(i-1)}\left(-1 + \log \frac{M_{i(i-1)}}{\hat{P}_{x^n}^*(i-1)}\right)$$

$$\gtrsim M_{i(i-1)}\left(-1 + \log\left(\frac{\log n}{8 \log \log n}\right)\right)$$

$$\asymp M_{i(i-1)} \log \log n.$$

By Lemma 5,

$$\Pr(X^n \in K_\ell(i)) \gtrsim \frac{k-1}{ek}\frac{1}{n}\left(1 - \frac{k-2}{n} - M_{i(i-1)}\right)^{\ell-1}.$$

Therefore,

$$\rho_n^{\mathrm{KL}}(\mathscr{P}) \geq \frac{1}{|\mathscr{P}_S|} \sum_{(M) \in \mathscr{P}_S} \sum_{\ell=\ell_1(M)}^{\ell_2(M)} \sum_{i \in [k]^e} \left[\Pr_{X^n \sim P}(X^n \in K_\ell(i)) D_{\mathrm{KL}}(P_{x^n}, \hat{P}_{x^n}^*; x^n \in K_\ell(i))\right]$$

$$\gtrsim \frac{(k-1)\log\log n}{enk} \sum_{i \in [k]^e} \frac{1}{|\mathscr{P}_S|} \sum_{\substack{(M) \in \mathscr{P}_S \\ \text{and } M_{i(i-1)} \in V_n'}} \sum_{\ell=\ell_1(M)}^{\ell_2(M)} \left(1 - \frac{k-2}{n} - M_{i(i-1)}\right)^{\ell-1} M_{i(i-1)}$$

$$\geq \frac{(k-1)\log\log n}{enk} \sum_{i \in [k]^e} \frac{1}{|V_n|} \sum_{v \in V_n'} \sum_{\ell=\frac{1}{v}\frac{1}{\log\log n}}^{\frac{1}{v}\log\log n} \left(1 - \frac{k-2}{n} - v\right)^{\ell-1} v,$$

where the last step follows by symmetry.

Next, we show that for any $v = \frac{1}{(\log n)^m} \in V_n'$,

$$T_m := \sum_{\ell=\frac{1}{v}\frac{1}{\log\log n}}^{\frac{1}{v}\log\log n} \left(1 - \frac{k-2}{n} - v\right)^{\ell-1} v \gtrsim 1.$$

Noting that $T_m$ is simply the summation of a geometric sequence, we can compute it as follows

$$T_m = \frac{1}{(\log n)^m} \sum_{\ell=\frac{(\log n)^m}{\log\log n}}^{(\log n)^m \log\log n} \left[\left(1 - \frac{k-2}{n} - \frac{1}{(\log n)^m}\right)^{\ell-1}\right]$$

$$= \frac{1}{(\log n)^m} \frac{\left(1 - \frac{k-2}{n} - \frac{1}{(\log n)^m}\right)^{\frac{(\log n)^m}{\log\log n}-1} - \left(1 - \frac{k-2}{n} - \frac{1}{(\log n)^m}\right)^{(\log n)^m \log\log n}}{1 - \left(1 - \frac{k-2}{n} - \frac{1}{(\log n)^m}\right)}$$

$$= \frac{1}{\frac{(k-2)(\log n)^m}{n} + 1}\left[\left(1 - \frac{k-2}{n} - \frac{1}{(\log n)^m}\right)^{\frac{(\log n)^m}{\log\log n}-1}\right.$$

$$\left. - \left(1 - \frac{k-2}{n} - \frac{1}{(\log n)^m}\right)^{(\log n)^m \log\log n}\right].$$

To provide a lower bound for $T_m$, we use the following inequalities:

$$\frac{1}{\frac{(k-2)(\log n)^m}{n} + 1} \geq \frac{1}{\frac{(k-2)(\log n)^{\frac{\log n}{4\log\log n}}}{n} + 1} = \frac{1}{\frac{(k-2)n^{\frac{1}{4}}}{n} + 1} \asymp 1,$$

$$\left(1-\frac{k-2}{n}-\frac{1}{(\log n)^m}\right)^{\frac{(\log n)^m}{\log\log n}-1} \geq \left[\left(1-\frac{k-2}{n}-\frac{1}{(\log n)^m}\right)^{\frac{1}{\frac{1}{(\log)^m}+\frac{k-2}{n}}}\right]^{\left(1+\frac{(k-2)(\log n)^m}{n}\right)\frac{1}{\log\log n}}$$

$$\geq \left(\frac{1}{4}\right)^{\left(1+\frac{(k-2)\sqrt{n}}{n}\right)\frac{1}{\log\log n}} \geq \left(\frac{1}{4}\right)^{2\frac{1}{\log\log n}} \asymp 1,$$

and

$$\left(1-\frac{k-2}{n}-\frac{1}{(\log n)^m}\right)^{(\log n)^m\log\log n}$$

$$= \left[\left(1-\frac{k-2}{n}-\frac{1}{(\log n)^m}\right)^{\frac{1}{\frac{k-2}{n}+\frac{1}{(\log n)^m}}}\right]^{\left(\frac{(k-2)(\log n)^m}{n}+1\right)\log\log n}$$

$$\leq \left(\frac{1}{e}\right)^{\log\log n} = \frac{1}{\log n}.$$

Consolidating these three inequalities, the sum $T_m$ can be lower bounded by

$$T_m \gtrsim 1(1-\frac{1}{\log n}) \asymp 1.$$

Finally,

$$\rho_n^{\mathrm{KL}}(\mathscr{P}) \gtrsim \frac{(k-1)\log\log n}{enk}\sum_{i\in[k]^e}\frac{1}{|V_n|}\sum_{v\in V_n'}(1-o(1))$$

$$= \frac{(k-1)\log\log n}{enk}\frac{k}{2}\frac{|V_n'|}{|V_n|}$$

$$= \frac{(k-1)\log\log n}{4en}.$$

## 2 Minimax prediction: upper bound

The proof makes use of the following lemma, which provides a uniform upper bound for the hitting probability of any $k$-state Markov chain.

**Lemma 6** *[1] For any Markov chain over $[k]$ and any two states $i,j\in[k]$, if $n>k$, then*

$$Pr_i(\tau(j)=n) \leq \frac{k}{n}.$$

Let $\mathscr{K}_n$ be the same as is in the previous section. Recall that

$$\rho_n^{\mathrm{KL}}(P,\hat{P};\mathscr{K}_n) = \sum_{x^n\in\mathscr{K}_n}P(x^n)D_{\mathrm{KL}}(P_{x^n},\hat{P}_{x^n}),$$

we denote the *partial minimax prediction risk over $\mathscr{K}_n$* by

$$\rho_n^{\mathrm{KL}}(\mathscr{P};\mathscr{K}_n) := \min_{\hat{P}}\max_{P\in\mathscr{P}}\rho_n^{\mathrm{KL}}(P,\hat{P};\mathscr{K}_n).$$

Let $\overline{\mathscr{K}_n} := [k]^n\setminus\mathscr{K}_n$, we define $\rho_n^{\mathrm{KL}}(P,\hat{P};\overline{\mathscr{K}_n})$ and $\rho_n^{\mathrm{KL}}(\mathscr{P};\overline{\mathscr{K}_n})$ in the same manner. As the consequence of $\hat{P}$ being a function from $[k]^n$ to $\Delta_k$, we have the following triangle inequality,

$$\rho_n^{\mathrm{KL}}(\mathscr{P}) \leq \rho_n^{\mathrm{KL}}(\mathscr{P};\overline{\mathscr{K}_n}) + \rho_n^{\mathrm{KL}}(\mathscr{P};\mathscr{K}_n).$$

Turning back to Markov chains, the next lemma upper bounds $\rho_n^{\mathrm{KL}}(\mathbb{M}^k;\overline{\mathscr{K}_n})$.

**Lemma 7** *Let $\hat{P}^{+\frac{1}{2}}$ denote the estimator that maps $X^n \sim (M)$ to $\hat{M}^{+\frac{1}{2}}(X_n, \cdot)$, then*

$$\max_{P \in \mathbb{M}^k} \rho_n^{KL}(P, \hat{P}^{+\frac{1}{2}}; \overline{\mathscr{K}_n}) \leq \mathcal{O}_k\left(\frac{1}{n}\right),$$

*which implies*

$$\rho_n^{KL}(\mathbb{M}^k; \overline{\mathscr{K}_n}) \leq \mathcal{O}_k\left(\frac{1}{n}\right).$$

**Proof**   The proof of this lemma is essentially a combination of the upper bounds' proofs in [2] and in Section 4. Instead of using the fact that $M_{ij}$ are bounded away from 0 (see Section 4), we partition $\overline{\mathscr{K}_n}$ into different subsets according to how close the counts are to their expected values, the number of times that the last appearing state transitioning to itself, and the number of times that the last appearing state transitioning to other states. Then, we bound the estimator's expected loss over each set of the partition by $\mathcal{O}_k(1/n)$. We omit the proof for the sake of brevity.

Recall the following lower bound,

$$\rho_n^{KL}(\mathbb{M}^k) = \Omega_k\left(\frac{\log \log n}{n}\right).$$

This together with Lemma 3 and the triangle inequality above shows that an upper bound on $\rho_n^{KL}(\mathbb{M}^k; \mathscr{K}_n)$ also suffices to bound the leading term of $\rho_n^{KL}(\mathbb{M}^k)$. The following lemma provides such an upper bound. Recall that for any $i \in [k]$, $K_\ell(i)$ is defined as $\{x^n \in [k]^n : x^n = \bar{i}^{n-\ell}i^\ell\}$.

**Lemma 8** *For any $x^n \in \mathscr{K}_n$, there exists a unique pair $(\ell, i)$ such that $x^n \in K_\ell(i)$. Consider the following estimator*

$$\hat{P}_{x^n}(i) := \begin{cases} 1 - \frac{1}{\ell \log n} & \ell \leq \frac{n}{2} \\ 1 - \frac{1}{\ell} & \ell > \frac{n}{2} \end{cases}$$

*and*

$$\hat{P}_{x^n}(j) := \frac{1 - \hat{P}_{x^n}(i)}{k - 1}, \ \forall j \in [k] \setminus \{i\},$$

*then we have*

$$\rho_n^{KL}(\mathbb{M}^k; \mathscr{K}_n) \leq \max_{P \in \mathbb{M}^k} \rho_n^{KL}(P, \hat{P}; \mathscr{K}_n) \lesssim \frac{2k^2 \log \log n}{n}.$$

**Proof**   Let $i \in [k]$ be an arbitrary state. For simplicity of illustration, we use the following notation: for any $x^n = \bar{i}^{n-\ell}i^\ell$, denote $\hat{p}_\ell := \hat{P}_{x^n}$; for any $(M) \in \mathbb{M}^k$, denote $p_i := M(i, \cdot)$; for any $\ell \leq n$, denote $h_{i,\ell} := \Pr(\tau(i) = \ell)$. By Lemma 6, the hitting probability $h_{i,\ell}$ is upper bounded by $k/\ell$ for all $\ell > k$. We can write

$$\rho_n^{KL}(P, \hat{P}; \mathscr{K}_n) = \sum_{i \in [k]} \sum_{\ell=1}^{n-1} h_{i,n-\ell}(p_i(i))^{\ell-1} D_{KL}(p_i, \hat{p}_\ell).$$

Now, we break the right hand side into two sums according to whether $\ell$ is greater than $n/2$ or not. For $\ell > n/2$, we have

$$\sum_{i\in[k]}\sum_{\ell=\frac{n}{2}+1}^{n-1} h_{i,n-\ell}(p_i(i))^{\ell-1}D_{\mathrm{KL}}(p_i,\hat{p}_\ell)$$

$$\leq \sum_{i\in[k]}\sum_{\ell=\frac{n}{2}+1}^{n-1} h_{i,n-\ell}(p_i(i))^{\ell-1}\left(p_i(i)\log\left(\frac{p_i(i)}{1-\frac{1}{\ell}}\right)+\sum_{j\neq i}p_i(j)\log\left(\frac{\sum_{j\neq i}p_i(j)}{\frac{1}{\ell(k-1)}}\right)\right)$$

$$\leq \sum_{i\in[k]}\sum_{\ell=\frac{n}{2}+1}^{n-1} h_{i,n-\ell}(p_i(i))^{\ell-1}\left(\log\left(\frac{1}{1-\frac{1}{\ell}}\right)+(1-p_i(i))\log\left(\ell(k-1)(1-p_i(i))\right)\right)$$

$$\leq \sum_{i\in[k]}\sum_{\ell=\frac{n}{2}+1}^{n-1} h_{i,n-\ell}(p_i(i))^{\ell-1}\left(\frac{\frac{1}{\ell}}{1-\frac{1}{\ell}}+(1-p_i(i))^2\ell(k-1)\right)$$

$$\leq \sum_{i\in[k]}\sum_{\ell=\frac{n}{2}+1}^{n-1} h_{i,n-\ell}\left(\frac{2}{n}+(p_i(i))^{\ell-1}(1-p_i(i))^2\ell(k-1)\right)$$

$$\leq \sum_{i\in[k]}\sum_{\ell=\frac{n}{2}+1}^{n-1} h_{i,n-\ell}\left(\frac{2}{n}+\frac{1}{(\ell+1)^2}\ell(k-1)\right)$$

$$\leq \sum_{i\in[k]}\sum_{\ell=\frac{n}{2}+1}^{n-1} h_{i,n-\ell}\left(\frac{2k}{n}\right)$$

$$= \sum_{i\in[k]}\frac{2k}{n}\Pr(\tau(i)\in[1,n/2-1])\leq \frac{2k^2}{n}.$$

Similarly, for $\ell \leq n/2$, we have

$$\sum_{i\in[k]}\sum_{\ell=1}^{\frac{n}{2}} h_{i,n-\ell}(p_i(i))^{\ell-1}D_{\mathrm{KL}}(p_i,\hat{p}_\ell)$$

$$\leq \sum_{i\in[k]}\sum_{\ell=1}^{\frac{n}{2}} h_{i,n-\ell}(p_i(i))^{\ell-1}\left(\log\left(\frac{1}{1-\frac{1}{\ell\log n}}\right)+(1-p_i(i))\log\left(\ell(k-1)(1-p_i(i))\log n\right)\right)$$

$$\leq \sum_{i\in[k]}\sum_{\ell=1}^{\frac{n}{2}} \frac{2k}{n}(p_i(i))^{\ell-1}\left(\frac{2}{\ell\log n}+(1-p_i(i))^2\ell(k-1)+(1-p_i(i))\log\log n\right)$$

$$\leq \sum_{i\in[k]}\frac{2k}{n}\left(\sum_{\ell=1}^{\frac{n}{2}}\frac{2}{\ell\log n}+\sum_{\ell=1}^{\frac{n}{2}}\ell(p_i(i))^{\ell-1}(1-p_i(i))^2(k-1)+\sum_{\ell=1}^{\frac{n}{2}}(p_i(i))^{\ell-1}(1-p_i(i))\log\log n\right)$$

$$\leq \sum_{i\in[k]}\frac{2k}{n}(2+(k-1)+\log\log n)$$

$$\asymp \frac{2k^2\log\log n}{n}.$$

This completes the proof.

# 3 Minimax estimation: lower bound

The proof of the lower bound makes use of the following concentration inequality, which upper bounds the probability that a binomial random variable exceeds its mean.

**Lemma 9** *[3] Let $Y$ be a binomial random variable with parameters $m \in \mathbb{N}$ and $p \in [0, 1]$, then for any $\epsilon \in (0, 1)$,*

$$Pr(Y \geq (1 + \epsilon)mp) \leq \exp\left(-\epsilon^2 mp/3\right).$$

### 3.1 Prior construction

Again we use the following standard argument to lower bound the minimax risk,

$$\varepsilon_n^L(\mathcal{M}) = \min_{\hat{M}} \max_{(M) \in \mathcal{M}} \varepsilon_n^L(M, \hat{M}) \geq \min_{\hat{M}} \mathbb{E}_{(M) \sim U(\mathcal{M}_S)}[\varepsilon_n^L(M, \hat{M})],$$

where $\mathcal{M}_S \subset \mathcal{M}$ and $U(\mathcal{M}_S)$ is the uniform distribution over $\mathcal{M}_S$. Setting $\mathcal{M} = \mathbb{M}_{\delta, \pi^*}^k$, we outline the construction of $\mathcal{M}_S$ as follows.

We adopt the notation in [4] and denote the $L_\infty$ ball of radius $r$ around $u_{k-1}$, the uniform distribution over $[k-1]$, by

$$B_{k-1}(r) := \{p \in \Delta_{k-1} : L_\infty(p, u_{k-1}) < r\},$$

where $L_\infty(\cdot, \cdot)$ is the $L_\infty$ distance between two distributions. For simplicity, define

$$p' := (p_1,\ p_2,\ \ldots,\ p_{k-1}),$$

$$p^* := \left(\frac{\bar{\pi}^*}{k-1},\ \frac{\bar{\pi}^*}{k-1},\ \cdots\ \frac{\bar{\pi}^*}{k-1},\ \pi^*\right),$$

and

$$M_n(p') := \begin{bmatrix} \frac{\bar{\pi}^*}{k-1} & \frac{\bar{\pi}^*}{k-1} & \cdots & \frac{\bar{\pi}^*}{k-1} & \pi^* \\ \frac{\bar{\pi}^*}{k-1} & \frac{\bar{\pi}^*}{k-1} & \cdots & \frac{\bar{\pi}^*}{k-1} & \pi^* \\ \vdots & \vdots & \ddots & \vdots & \vdots \\ \frac{\bar{\pi}^*}{k-1} & \frac{\bar{\pi}^*}{k-1} & \cdots & \frac{\bar{\pi}^*}{k-1} & \pi^* \\ \bar{\pi}^* p_1 & \bar{\pi}^* p_2 & \cdots & \bar{\pi}^* p_{k-1} & \pi^* \end{bmatrix},$$

where $\bar{\pi}^* = 1 - \pi^*$ and $\sum_{i=1}^{k-1} p_i = 1$.

Given $n$ and $\epsilon \in (0, 1)$, let $n' := (n(1 + \epsilon)\pi^*)^{1/5}$. We set

$$\mathcal{M}_S = \{(M) \in \mathbb{M}_{\delta, \pi^*}^k : \mu = p^* \text{ and } M = M_n(p'), \text{ where } p' \in B_{k-1}(1/n')\}.$$

Noting that the uniform distribution over $\mathcal{M}_S$, $U(\mathcal{M}_S)$, is induced by $U(B_{k-1}(1/n'))$, the uniform distribution over $B_{k-1}(1/n')$ and thus is well-defined.

An important property of the above construction is that for a sample sequence $X^n \sim (M) \in \mathcal{M}_S$, $N_k$, the number of times that state $k$ appears in $X^n$, is a binomial random variable with parameters $n$ and $\pi^*$. Therefore, Lemma 9 implies that $N_k$ is highly concentrated around its mean $n\pi^*$.

### 3.2 $L_2$-divergence lower bound

Let us first consider the $L_2$-distance. Similar to Lemma 1, $\hat{M}^*$, the estimator that minimizes $\mathbb{E}_{(M) \sim U(\mathcal{M}_S)}[\varepsilon_n^{L_2}(M, \hat{M})]$, can be computed exactly. In particular, we have the following lemma.

**Lemma 10** *There exists an estimator $\hat{M}^*$ with*

$$\hat{M}_{x^n}^*(i, \cdot) = p^*, \forall i \in [k - 1],$$

*and*

$$\hat{M}_{x^n}^*(k, k) = \pi^*,$$

*such that $\hat{M}^*$ minimizes $\mathbb{E}_{(M) \sim U(\mathcal{M}_S)}[\varepsilon_n^{L_2}(M, \hat{M})]$.*

Based on the above lemma, we can relate the minimax estimation risk of Markov chains to the minimax prediction risk of *i.i.d.* processes. For simplicity, denote $\mathcal{B}_{i.i.d.} := \{(p) \in \mathbb{IID}^{k-1} : p \in B_{k-1}(1/n')\}$. The following lemma holds.

**Lemma 11** *For any $x^n \in [k]^n$, let $\mathbb{I}(x^n)$ be the collection of indexes $j \in [n]$ such that $x_j = k$. Then,*

$$\mathbb{E}_{(M) \sim U(\mathscr{M}_S)}[\mathbb{E}_{X^n \sim (M)}[L_2(M(k, \cdot), \hat{M}_{X^n}^*(k, \cdot))\mathbb{1}_{\mathbb{I}(X^n) = \mathbb{I}_0}]]$$
$$= C(\mathbb{I}_0, \pi^*, p^*, n) \min_{\hat{P}} \mathbb{E}_{P \sim U(\mathscr{B}_{i.i.d.})}[\rho_{|\mathbb{I}_0|}^{L_2}(P, \hat{P})],$$

*where $\mathbb{I}_0$ is an arbitrary non-empty subset of $[n]$ and $C(\mathbb{I}_0, \pi^*, p^*, n)$ is a constant whose value only depends on $\mathbb{I}_0, \pi^*, p^*$, and $n$.*

**Proof**    We first consider the inner expectation on the left-hand side of the equality. For any $(M) \in \mathscr{M}_S$, we have

$$\mathbb{E}_{X^n \sim (M)}[L_2(M(k, \cdot), \hat{M}_{X^n}^*(k, \cdot))\mathbb{1}_{\mathbb{I}(X^n) = \mathbb{I}_0}]$$
$$= \sum_{x^n : \mathbb{I}(x^n) = \mathbb{I}_0} P(x^n) L_2(M(k, \cdot), \hat{M}_{x^n}^*(k, \cdot))$$
$$= \sum_{x^n : \mathbb{I}(x^n) = \mathbb{I}_0} \mu(x_1) \prod_{t=1}^{n-1} M(x_t, x_{t+1}) L_2(M(k, \cdot), \hat{M}_{x^n}^*(k, \cdot)).$$

Let us partition $\mathbb{I}_0$ into two parts: the collection of indexes $m \in \mathbb{I}_0 \cap [n-1]$ such that $m \in \mathbb{I}_0$ and $m + 1 \notin \mathbb{I}_0$, say $\{m_1, \ldots, m_s\}$, and the remaining elements in $\mathbb{I}_0$. By the construction of $\mathscr{M}_S$, we have

$$\sum_{x^n : \mathbb{I}(x^n) = \mathbb{I}_0} \mu(x_1) \prod_{t=1}^{n-1} M(x_t, x_{t+1}) L_2(M(k, \cdot), \hat{M}_{x^n}^*(k, \cdot))$$
$$= (\pi^*)^{|\mathbb{I}_0|} \left( \frac{\bar{\pi}^*}{k-1} \right)^{n-s-|\mathbb{I}_0|} \sum_{x^n : \mathbb{I}(x^n) = \mathbb{I}_0} \prod_{t=1}^{s} M(k, x_{m_t+1}) L_2(M(k, \cdot), \hat{M}_{x^n}^*(k, \cdot)).$$

For any $x^n$, let $x^n \setminus \mathbb{I}_0$ denote the subsequence $x_{j_1}, \ldots, x_{j_{n-|\mathbb{I}_0|-s}}$ such that $j_1 < j_2 \ldots < j_{n-|\mathbb{I}_0|-s}$, $j_t \notin \mathbb{I}_0$ and $j_t - 1 \notin \{m_1, \ldots, m_s\}, \forall t$. We can further partition the last summation according to $x^n \setminus \mathbb{I}_0$ as follows.

$$\sum_{x^n : \mathbb{I}(x^n) = \mathbb{I}_0} \prod_{t=1}^{s} M(k, x_{m_t+1}) L_2(M(k, \cdot), \hat{M}_{x^n}^*(k, \cdot))$$
$$= \sum_{y^{n-|\mathbb{I}_0|-s} \in [k-1]^{n-|\mathbb{I}_0|-s}} \left( \sum_{\substack{x^n : x_j = k, \forall j \in \mathbb{I}_0 \\ \text{and } x^n \setminus \mathbb{I}_0 = y^{n-|\mathbb{I}_0|-s}}} \prod_{t=1}^{s} M(k, x_{m_t+1}) L_2(M(k, \cdot), \hat{M}_{x^n}^*(k, \cdot)) \right).$$

Fixing $y^{n-|\mathbb{I}_0|-s} \in [k-1]^{n-|\mathbb{I}_0|-s}$, there is a bijective mapping from $S(\mathbb{I}_0, y^{n-|\mathbb{I}_0|-s}) := \{x^n : x_j = k, \forall j \in \mathbb{I}_0 \text{ and } x^n \setminus \mathbb{I}_0 = y^{n-|\mathbb{I}_0|-s}\}$ to $[k-1]^s$, say $g(\cdot)$. Furthermore, we have $\hat{M}^*(k, k) = \pi^*$. Hence, we can denote $q_{g(x^n)}^* := \frac{\hat{M}_{x^n}^*(k, [k-1])}{\bar{\pi}^*}$ for $x^n \in S(\mathbb{I}_0, y^{n-|\mathbb{I}_0|-s})$ and treat it as a mapping from $[k-1]^s$ to $\Delta_{k-1}$. Also, $(M) \in \mathscr{M}_S$ implies that $M(k, [k-1]) = p'$ for some $p' \in B_{k-1}(1/n')$. Thus,

$$L_2(M(k, \cdot), \hat{M}_{x^n}^*(k, \cdot)) = (\bar{\pi}^*)^2 L_2(p', q_{g(x^n)}^*),$$

$$\prod_{t=1}^{s} M(k, x_{m_t+1}) L_2(M(k, \cdot), \hat{M}_{x^n}^*(k, \cdot)) = \prod_{t=1}^{s} p'(x_{m_t+1})(\bar{\pi}^*)^2 L_2(p', q_{g(x^n)}^*),$$

and

$$\sum_{x^n \in S(\mathbb{I}_0, y^{n-|\mathbb{I}_0|-s})} \prod_{t=1}^{s} M(k, x_{m_t+1}) L_2(M(k, \cdot), \hat{M}_{x^n}^*(k, \cdot))$$

$$= \sum_{x^n \in S(\mathbb{I}_0, y^{n-|\mathbb{I}_0|-s})} \prod_{t=1}^{s} p'(x_{m_t+1})(\bar{\pi}^*)^2 L_2(p', q_{g(x^n)}^*)$$

$$= \sum_{z^s \in [k-1]^s} \prod_{t=1}^{s} p'(z_t)(\bar{\pi}^*)^2 L_2(p', q_{z^s}^*)$$

$$= \mathbb{E}_{Z^s \sim (p')}[(\bar{\pi}^*)^2 L_2(p', q_{Z^s}^*)],$$

where $(p')$ is an *i.i.d.* process whose underlying distribution is $p'$.

By definition, $\hat{M}^*$ minimizes $\mathbb{E}_{(M) \sim U(\mathcal{M}_S)}[\varepsilon_n^{L_2}(M, \hat{M})]$ and for each $x^n \in [k]^n$, its value $\hat{M}_{x^n}^*$ is completely determined by $x^n$. Besides, $\{S(\mathbb{I}_0, y^{n-|\mathbb{I}_0|-s}) : \mathbb{I}_0 \subset [n] \text{ and } y^{n-|\mathbb{I}_0|-s} \in [k-1]^{n-|\mathbb{I}_0|-s}\}$ forms a partition of $[k]^n$. Therefore, by the linearity of expectation and the definition of $q^*$, the estimator $q^*$ also minimizes $\mathbb{E}_{p' \sim U(B_{k-1}(1/n'))}[\mathbb{E}_{Z^s \sim (p')}[(\bar{\pi}^*)^2 L_2(p', q_{Z^s})]]$, where the minimization is over all the possible mappings $q$ from $[k-1]^s$ to $\Delta_{k-1}$. Equivalently, we have

$$\mathbb{E}_{p' \sim U(B_{k-1}(1/n'))}[\mathbb{E}_{Z^n \sim (p')}[(\bar{\pi}^*)^2 L_2(p', q_{Z^n}^*)]] = \min_{\hat{P}} \mathbb{E}_{P \sim U(\mathcal{B}_{i.i.d.})}(\bar{\pi}^*)^2[\rho_s^{L_2}(P, \hat{P})].$$

This immediately yields the lemma.

For any $(M) \in \mathcal{M}_S$, denote by $N_k((M), n)$ the number of times that state $k$ appears in $X^n \sim (M)$, which is a random variable induced by $(M)$ and $n$. Lemma 11, we can deduce that

**Lemma 12**

$$\min_{\hat{M}} \mathbb{E}_{(M) \sim U(\mathcal{M}_S)}[\varepsilon_n^{L_2}(M, \hat{M})] \geq \mathbb{E}_{(M) \sim U(\mathcal{M}_S)} \left[ (\bar{\pi}^*)^2 \min_{\hat{P}} \mathbb{E}_{P' \sim U(\mathcal{B}_{i.i.d})}[\rho_{N_k((M),n)}^{L_2}(P', \hat{P})] \right].$$

By Lemma 9 and our construction of $\mathcal{M}_S$, the probability that $N_k((M), n) \geq (1+\epsilon)n\pi^*$ is at most $\exp(-\epsilon^2 n\pi^*/3)$ for any $(M) \in \mathcal{M}_S$ and $\epsilon \in (0, 1)$. This together with Lemma 12 and

$$\min_{\hat{P}} \mathbb{E}_{P \sim U(\mathcal{B}_{i.i.d.})}[\rho_m^{L_2}(P, \hat{P})] \gtrsim \frac{1 - \frac{1}{k-1}}{(1+\epsilon)n\pi^*}, \forall m < (1+\epsilon)n\pi^*,$$

from [4] yields

**Lemma 13** *For all $\epsilon \in (0, 1)$,*

$$\varepsilon_n^{L_2}(\mathcal{M}) = \varepsilon_n^{L_2}(\mathbb{M}_{\delta, \pi^*}^k) \gtrsim \frac{(1 - \frac{1}{k-1})(1 - \pi^*)^2}{n\pi^*(1+\epsilon)}.$$

### 3.3 Lower bound for ordinary $f$-divergences

Now we proceed from the $L_2$-distance to ordinary $f$-divergences. The following lemma from [4] shows that $D_f(p, q)$ decreases if we move $q$ closer to $p$.

**Lemma 14** *For $p_1 > q_1$, $p_2 < q_2$ and $d \leq \min\{p_1 - q_1, q_2 - p_2\}$,*

$$q_1 f\left(\frac{p_1}{q_1}\right) + q_2 f\left(\frac{p_2}{q_2}\right) \geq (q_1 + d) f\left(\frac{p_1}{q_1 + d}\right) + (q_2 - d) f\left(\frac{p_2}{q_2 - d}\right).$$

Based on the above lemma, we show that for any $x^n \in [k]^n$, the value of the optimal estimator is always close to $(u_{k-1}\bar{\pi}^*, \pi^*)$.

Let $\hat{p}_{x^n}^* := \hat{M}_{x^n}^*(k, \cdot)$. For any $x^n \in [k]^n$, we claim that either $\hat{p}_{x^n}^*(j) \geq (\frac{1}{k-1} - \frac{1}{n'})\bar{\pi}^*, \forall j \in [k-1]$ and $\hat{p}_{x^n}^*(k) \geq \pi^*$ OR $\hat{p}_{x^n}^*(j) \leq (\frac{1}{k-1} + \frac{1}{n'})\bar{\pi}^*, \forall j \in [k-1]$ and $\hat{p}_{x^n}^*(k) \leq \pi^*$. Otherwise, Lemma 14 implies that we can reduce the estimation risk by moving $\hat{p}_{x^n}^*$ closer to $(u_{k-1}\bar{\pi}^*, \pi^*)$.

If $\hat{p}^*_{x^n}(j) \geq (\frac{1}{k-1} - \frac{1}{n'})\bar{\pi}^*$, $\forall j \in [k-1]$ and $\hat{p}^*_{x^n}(k) \geq \pi^*$, then $\hat{p}^*_{x^n}(j) \leq (\frac{1}{k-1} + \frac{k-2}{n'})\bar{\pi}^*$, $\forall j \in [k-1]$ and $\hat{p}^*_{x^n}(k) \leq \pi^* + \frac{k-1}{n'}\bar{\pi}^*$. Similarly, if $\hat{p}^*_{x^n}(j) \leq (\frac{1}{k-1} + \frac{1}{n'})\bar{\pi}^*$, $\forall j \in [k-1]$ and $\hat{p}^*_{x^n}(k) \leq \pi^*$, then $\hat{p}^*_{x^n}(j) \geq (\frac{1}{k-1} - \frac{k-2}{n'})\bar{\pi}^*$, $\forall j \in [k-1]$ and $\hat{p}^*_{x^n}(k) \geq \pi^* - \frac{k-1}{n'}\bar{\pi}^*$.

Now we relate $D_f(p, \hat{p}^*)$ to $L_2(p, \hat{p}^*)$. For simplicity, denote $p := M(k, \cdot)$ and drop $x^n$ from $\hat{p}^*_{x^n}$.

**Lemma 15** *For sufficiently large $n$,*

$$D_f(p, \hat{p}^*) \asymp \frac{(k-1)f''(1)}{2} L_2(p, \hat{p}^*).$$

**Proof**  By the previous lemma, $\hat{p}^*_{x^n}(j) = (\frac{1}{k-1} \pm \frac{k-2}{n'})\bar{\pi}^*$, $\forall j \in [k-1]$ and $\hat{p}^*_{x^n}(k) = \pi^* \pm \frac{k-1}{n'}\bar{\pi}^*$. Therefore,

$$\frac{p(i)}{\hat{p}^*(i)} \in \left[ \frac{n' - \frac{k}{\delta}}{n' + \frac{k}{\delta}}, \frac{n' + \frac{k}{\delta}}{n' - \frac{k}{\delta}} \right], \forall i \in [k].$$

Let us denote the interval on the right hand side by $I$.

For sufficiently large $n$, we can apply the second-order Taylor expansion to $f$ at point 1.

$$
\begin{aligned}
D_f(p, \hat{p}^*) &= \sum_{i \in [k]} \hat{p}^*(i) f\left( \frac{p(i)}{\hat{p}^*(i)} \right) \\
&= \sum_{i \in [k]} \left( \hat{p}^*(i)\left( \frac{p(i)}{\hat{p}^*(i)} - 1 \right) f'(1) + \frac{\hat{p}^*(i)}{2}\left( \frac{p(i)}{\hat{p}^*(i)} - 1 \right)^2 f''(1) \right.\\
&\quad \left. \pm \frac{\hat{p}^*(i)}{6}\left| \frac{p(i)}{\hat{p}^*(i)} - 1 \right|^3 \max_{z \in I}|f'''(z)| \right) \\
&= \sum_{i \in [k]} \left( \frac{\hat{p}^*(i)}{2}\left( \frac{p(i)}{\hat{p}^*(i)} - 1 \right)^2 f''(1) \pm \frac{1}{6}\frac{k}{n'}\left( \frac{p(i)}{\hat{p}^*(i)} - 1 \right)^2 \max_{z \in I}|f'''(z)| \right) \\
&\gtrsim \frac{f''(1)}{2} \sum_{i \in [k-1]} \hat{p}^*(i)\left( \frac{p(i)}{\hat{p}^*(i)} - 1 \right)^2 \\
&\asymp \frac{(k-1)f''(1)}{2\bar{\pi}^*} L_2(p, \hat{p}^*).
\end{aligned}
$$

Lemma 15 together with Lemma 13 yields

**Lemma 16** *For sufficiently large $n$,*

$$\varepsilon_n^f(\mathbb{M}^k_{\delta, \pi^*}) \gtrsim (1 - \pi^*)\frac{(k-2)f''(1)}{2n\pi^*}.$$

## 4  Minimax estimation: upper bound

### 4.1  Concentration of the counts

The proof of the upper bound relies on the following concentration inequality, which shows that for any Markov chain in $\mathbb{M}^k_\delta$ and any state $i \in [k]$, with high probability $N_i$ stays close to $(n-1)\pi_i$, for sufficiently large $n$.

**Lemma 17** *Given a sample sequence $X^n$ from any Markov chain $(M) \in \mathbb{M}^k_\delta$, let $N_i$ denote the number of times that symbol $i$ appears in $X^{n-1}$. Then for any $t \geq 0$,*

$$Pr(|N_i - (n-1)\pi_i| > t) \leq \sqrt{\frac{2}{\delta}} \exp\left( \frac{-t^2/C(\delta)}{4((n-1) + 2C(\delta)) + 40t} \right),$$

where $\pi$ is the stationary distribution of $(M)$ and

$$C(\delta) := \left\lceil -\frac{\ln 4}{\ln (1 - \delta)} + 1 \right\rceil.$$

**Proof**    Given $(M) \in \mathbb{M}_\delta^k$, recall that $P^{n+1}$ denotes the distribution of $X_{n+1}$ if we draw $X^{n+1} \sim (M)$. First, we show that

$$D_{L_1}(P^{n+1}, \pi) \leq 2(1 - \delta)^n.$$

Let $\Pi$ be the $k \times k$ matrix such that $\Pi(i, \cdot) = \pi$ for all $i \in [k]$. Noting that $M(i, j) \geq \delta \Pi(i, j)$, we can define

$$M_\delta := \frac{M - \delta \Pi}{1 - \delta},$$

which is also a valid transition matrix.

By induction, we can show

$$M^n = (1 - (1 - \delta)^n)\Pi + (1 - \delta)^n M_\delta^n.$$

Let us rearrange the terms:

$$M^n - \Pi = (1 - \delta)^n (M_\delta^n - \Pi).$$

Hence, let $|\cdot|$ denote the $L_1$ norm, we have

$$D_{L_1}(P^{n+1}, \pi) = |\mu(M^n - \Pi)| = |(1 - \delta)^n \mu(M_\delta^n - \Pi)| \leq 2(1 - \delta)^n.$$

This implies that we can upper bound $t_{mix}$ by $C(\delta)$.

The remaining proof follows from Proposition 3.4, Theorem 3.4, and Proposition 3.10 of [5] and is omitted here for the sake of brevity.

Noting that $\Pr(|N_i - (n - 1)\pi_i| > (n - 1)) = 0$, we have

$$\Pr(|N_i - (n - 1)\pi_i| > t) \leq \sqrt{\frac{2}{\delta}} \exp\left(\frac{-t^2}{4C(\delta)(11(n - 1) + 2C(\delta))}\right).$$

Informally, we can express the above inequality as

$$\Pr(|N_i - (n - 1)\pi_i| > t) \leq \Theta_\delta(\exp(\Theta_\delta(-t^2/n))),$$

which is very similar to the Hoeffding's inequality for the *i.i.d.* processes. As an important implication, the following lemma bounds the moments of $|N_i - (n - 1)\pi_i|$.

**Lemma 18**  *For $N_i$ defined in Lemma 17 and any $m \in \mathbb{Z}^+$,*

$$\mathbb{E}[|N_i - (n - 1)\pi_i|^m] \leq \frac{m\Gamma(m/2)}{\sqrt{2\delta}}(4C(\delta)(11(n - 1) + 2C(\delta)))^{m/2}.$$

**Proof**    The statement follows from

$$\mathbb{E}[|N_i - (n - 1)\pi_i|^m] = \int_0^\infty \Pr(|N_i - (n - 1)\pi_i|^m > t)\, dt$$

$$= \int_0^\infty \Pr(|N_i - (n - 1)\pi_i| > t^{1/m})\, dt$$

$$\leq \sqrt{\frac{2}{\delta}} \int_0^\infty \exp\left(\frac{-t^{2/m}}{4C(\delta)(11n + 2C(\delta))}\right) dt$$

$$= \frac{m}{\sqrt{2\delta}}(4C(\delta)(11n + 2C(\delta)))^{m/2} \int_0^\infty e^{-y} y^{m/2 - 1}\, dy$$

$$= \frac{m\Gamma(m/2)}{\sqrt{2\delta}}(4C(\delta)(11n + 2C(\delta)))^{m/2}.$$

## 4.2 A modified add-$\beta$ estimator

The difficulty with analyzing the performance of the original add-$\beta$ estimator is that the chain's initial distribution could be far away from its stationary distribution and finding a simple expression for $\mathbb{E}[N_i]$ and $\mathbb{E}[N_{ij}]$ could be hard. To overcome such difficulty, we ignore the first few sample points and construct a new add-$\beta$ estimator based on the remaining sample points. To be more specific, let $X^n$ be a length-$n$ sample sequence drawn from the Markov chain $(M)$. Removing the first $m$ sample points, $X_{m+1}^n := X_{m+1}, \ldots, X_n$ can be viewed as a length-$(n-m)$ sample sequence drawn from $(M)$ whose initial distribution $\mu'$ satisfies

$$L_1(\mu', \pi) < 2(1 - \delta)^{m-1}.$$

Setting $m = \sqrt{n}$, we have $L_1(\mu', \pi) \lesssim 1/n^2$. Noting that $\sqrt{n} \ll n$ for sufficiently large $n$, without loss of generality, we assume that the original distribution $\mu$ already satisfies $L_1(\mu, \pi) < 1/n^2$. If not, we can simply replace $X^n$ by $X_{\sqrt{n}+1}^n$.

To prove the upper bound, we consider the following (modified) add-$\beta$ estimator:

$$\hat{M}_{X^n}^{+\beta}(i, j) := \frac{N_{ij} + \beta}{N_i + k\beta}, \ \forall i, j \in [k],$$

where $\beta > 0$ is a fixed constant.

We can compute the expected values of these counts as

$$\mathbb{E}[N_i] = (n-1)\pi_i + \sum_{t=1}^{n-1} (\mathbb{E}[\mathbb{1}_{X_t=i}] - \pi_i)$$
$$= (n-1)\pi_i \pm \mathcal{O}(1/(n^2\delta))$$

and

$$\mathbb{E}[N_{ij}] = (n-1)\pi_i M_{ij} + \sum_{t=1}^{n-1} (\mathbb{E}[\mathbb{1}_{X_t=i}\mathbb{1}_{X_{t+1}=j}] - \pi_i M_{ij})$$
$$= (n-1)\pi_i M_{ij} + \sum_{t=1}^{n-1} (\mathbb{E}[\mathbb{1}_{X_t=i}] - \pi_i) M_{ij}$$
$$= (n-1)\pi_i M_{ij} \pm \mathcal{O}(1/(n^2\delta)).$$

## 4.3 Analysis

For notational convenience, let us re-denote $n' := n - 1$.

By Lemma 17,
$$\Pr\left(|N_i - n'\pi_i| > t\right) \le \Theta_\delta(\exp(\Theta_\delta(-t^2/n)))$$
and
$$\Pr\left(|N_{ij} - n'\pi_i M_{ij}| > t\right) \le \Theta_\delta(\exp(\Theta_\delta(-t^2/n))).$$

The second inequality follows from the fact that $N_{ij}$ can be viewed as the sum of counts from the following two Markov chains over $[k] \times [k]$ whose transition probabilities are greater than $\delta^2$:

$$(X_1, X_2), (X_3, X_4), \ldots$$

and

$$(X_2, X_3), (X_4, X_5), \ldots.$$

In other words, $N_i$ and $N_{ij}$ are highly concentrated around $n'\pi_i$ and $n'\pi_i M_{ij}$, respectively. Let $A_i$ denote the event that $N_i = n'\pi_i(1 \pm \delta/2)$ and $N_{ij} = n'\pi_i M_{ij}(1 \pm \delta/2)$, $\forall j \in [k]$. Let $A_i^C$ denote the event that $A_i$ does not happen. Applying the union bound, we have

$$\mathbb{E}[\mathbb{1}_{A_i^C}] = \Pr(A_i^C) \le \Theta_\delta(\exp(\Theta_\delta(-n))).$$

Now consider

$$D_f(p, q) = \sum_{i \in [k]} q(i) f\left(\frac{p(i)}{q(i)}\right),$$

the corresponding estimation risk of $\hat{M}^{+\beta}$ over a particular sate $i \in [k]$ can be decomposed as

$$\mathbb{E}[D_f(M(i, \cdot), \hat{M}^{+\beta}_{X^n}(i, \cdot)) \mathbb{1}_{A_i}] + \mathbb{E}[D_f(M(i, \cdot), \hat{M}^{+\beta}_{X^n}(i, \cdot)) \mathbb{1}_{A_i^C}].$$

Noting that

$$\hat{M}^{+\beta}_{X^n}(i, j) = \frac{N_{ij} + \beta}{N_i + k\beta} \in \left[\frac{\beta}{n + k\beta}, 1\right]$$

and $M_{ij} \in [\delta, 1]$, we have

$$|D_f(M(i, \cdot), \hat{M}^{+\beta}_{X^n}(i, \cdot))| \leq k \cdot \frac{n + \beta}{k\beta} \cdot \max_{y \in [\delta, k + n/\beta]} f(y).$$

Hence, we can bound the second term as

$$\begin{aligned}
\mathbb{E}[D_f(M(i, \cdot), \hat{M}^{+\beta}_{X^n}(i, \cdot)) \mathbb{1}_{A_i^C}] &\leq \frac{n + \beta}{\beta} \cdot \max_{y \in [\delta, k + n/\beta]} f(y) \cdot \mathbb{E}[\mathbb{1}_{A_i^C}] \\
&\leq \frac{n + \beta}{\beta} \cdot \max_{y \in [\delta, k + n/\beta]} f(y) \cdot \Theta_\delta(\exp(\Theta_\delta(-n))) \\
&= \frac{o(1)}{n},
\end{aligned}$$

where the last step follows from our assumption that $f$ is sub-exponential.

By the definition of $D_f$ and $\hat{M}^{+\beta}$,

$$\mathbb{E}\left[D_f(M(i, \cdot), \hat{M}^{+\beta}_{X^n}(i, \cdot)) \mathbb{1}_{A_i}\right] = \mathbb{E}\left[\sum_{j \in [k]} \frac{N_{ij} + \beta}{N_i + k\beta} f\left(\frac{M_{ij}}{\frac{N_{ij} + \beta}{N_i + k\beta}}\right) \mathbb{1}_{A_i}\right].$$

Let $h(x) := f\left(\frac{1}{1+x}\right)$, then $h$ is thrice continuously differentiable around some neighborhood of point 0 and

$$f(x) = h\left(\frac{1}{x} - 1\right).$$

We apply Taylor expansion to $h$ at point 0 and rewrite the expectation on the right-hand side as

$$\begin{aligned}
\mathbb{E} \sum_{j \in [k]} \frac{N_{ij} + \beta}{N_i + k\beta} f\left(\frac{M_{ij}}{\frac{N_{ij} + \beta}{N_i + k\beta}}\right) \mathbb{1}_{A_i} &= \mathbb{E} \sum_{j \in [k]} \frac{N_{ij} + \beta}{N_i + k\beta} h\left(\frac{(N_{ij} - M_{ij}N_i) + \beta(1 - kM_{ij})}{M_{ij}(N_i + k\beta)}\right) \mathbb{1}_{A_i} \\
&= \mathbb{E} \sum_{j \in [k]} \frac{N_{ij} + \beta}{N_i + k\beta} \left[h'(0) \frac{(N_{ij} - M_{ij}N_i) + \beta(1 - kM_{ij})}{M_{ij}(N_i + k\beta)} \right. \\
&\quad + \frac{h''(0)}{2}\left(\frac{(N_{ij} - M_{ij}N_i) + \beta(1 - kM_{ij})}{M_{ij}(N_i + k\beta)}\right)^2 \\
&\quad \left. \pm \frac{M(\delta)}{6}\left|\frac{(N_{ij} - M_{ij}N_i) + \beta(1 - kM_{ij})}{M_{ij}(N_i + k\beta)}\right|^3\right] \mathbb{1}_{A_i},
\end{aligned}$$

where by our definition of $A_i$, we set

$$M(\delta) := \max_{z \in \left[-\frac{2\delta}{1-\delta}, \frac{2\delta}{1-\delta}\right]} |h'''(z)|.$$

Now, we bound individual terms. Taking out $h'(0)$, the first term evaluates to:

$$\mathbb{E} \sum_{j\in[k]} \frac{N_{ij}+\beta}{N_i+k\beta} \frac{(N_{ij}-M_{ij}N_i)+\beta(1-kM_{ij})}{M_{ij}(N_i+k\beta)} \mathbb{1}_{A_i}$$

$$= \mathbb{E} \sum_{j\in[k]} ((N_{ij}-n'\pi_i M_{ij})+(n'\pi_i M_{ij}+\beta)) \frac{(N_{ij}-M_{ij}N_i)+\beta(1-kM_{ij})}{M_{ij}(N_i+k\beta)^2} \mathbb{1}_{A_i}$$

$$= \mathbb{E} \sum_{j\in[k]} \frac{(N_{ij}-n'\pi_i M_{ij})}{M_{ij}} \frac{N_{ij}-n'\pi_i M_{ij})+(n'\pi_i M_{ij}-M_{ij}N_i)+\beta(1-kM_{ij})}{(N_i+k\beta)^2} \mathbb{1}_{A_i}$$

$$+ \frac{(n'\pi_i M_{ij}+\beta)}{M_{ij}} \frac{(N_{ij}-M_{ij}N_i)+\beta(1-kM_{ij})}{(N_i+k\beta)^2} \mathbb{1}_{A_i}$$

$$= \mathbb{E} \sum_{j\in[k]} \frac{(N_{ij}-n'\pi_i M_{ij})}{M_{ij}} \frac{(N_{ij}-n'\pi_i M_{ij})}{(N_i+k\beta)^2} + \frac{(N_{ij}-n'\pi_i M_{ij})(n'\pi_i-N_i)}{(N_i+k\beta)^2}$$

$$+ n'\pi_i \frac{(N_{ij}-M_{ij}N_i)+\beta(1-kM_{ij})}{(N_i+k\beta)^2} + \frac{o(1)}{n}$$

$$= -\mathbb{E}\frac{(N_i-n'\pi_i)^2}{(N_i+k\beta)^2} + \mathbb{E} \sum_{j\in[k]} \frac{1}{M_{ij}} \frac{(N_{ij}-n'\pi_i M_{ij})^2}{(N_i+k\beta)^2} + \frac{o(1)}{n}$$

$$= -\mathbb{E}\frac{(N_i-n'\pi_i)^2}{(n'\pi_i+k\beta)^2} + \mathbb{E} \sum_{j\in[k]} \frac{1}{M_{ij}} \frac{(N_{ij}-n'\pi_i M_{ij})^2}{(n'\pi_i+k\beta)^2} + \frac{o(1)}{n}.$$

Taking out $h''(0)/2$, the second term evaluates to:

$$\mathbb{E} \sum_{j\in[k]} \frac{N_{ij}+\beta}{N_i+k\beta} \left( \frac{(N_{ij}-M_{ij}N_i)+\beta(1-kM_{ij})}{M_{ij}(N_i+k\beta)} \right)^2 \mathbb{1}_{A_i}$$

$$= \mathbb{E} \sum_{j\in[k]} ((N_{ij}-M_{ij}N_i)+(M_{ij}N_i+\beta)) \frac{((N_{ij}-M_{ij}N_i)+\beta(1-kM_{ij}))^2}{M_{ij}^2(N_i+k\beta)^3} \mathbb{1}_{A_i}$$

$$= \mathbb{E} \sum_{j\in[k]} (N_{ij}-M_{ij}N_i) \frac{((N_{ij}-M_{ij}N_i)+\beta(1-kM_{ij}))^2}{M_{ij}^2(N_i+k\beta)^3} \mathbb{1}_{A_i}$$

$$+ (M_{ij}N_i+\beta) \frac{((N_{ij}-M_{ij}N_i)+\beta(1-kM_{ij}))^2}{M_{ij}^2(N_i+k\beta)^3} \mathbb{1}_{A_i}$$

$$= \mathbb{E} \sum_{j\in[k]} (M_{ij}N_i+\beta) \frac{((N_{ij}-M_{ij}N_i)+\beta(1-kM_{ij}))^2}{M_{ij}^2(N_i+k\beta)^3} + \frac{o(1)}{n}$$

$$= \mathbb{E} \sum_{j\in[k]} \frac{1}{M_{ij}} \frac{(N_{ij}-M_{ij}N_i)^2}{(N_i+k\beta)^2} + 2\mathbb{E} \sum_{j\in[k]} (M_{ij}N_i+\beta) \frac{(N_{ij}-M_{ij}N_i)\beta(1-kM_{ij})}{M_{ij}^2(N_i+k\beta)^3} + \frac{o(1)}{n}$$

$$= \mathbb{E} \sum_{j\in[k]} \frac{1}{M_{ij}} \frac{(N_{ij}-n'M_{ij}\pi_i+n'M_{ij}\pi_i-M_{ij}N_i)^2}{(N_i+k\beta)^2} + \frac{o(1)}{n}$$

$$= -\mathbb{E}\frac{(N_i-n'\pi_i)^2}{(N_i+k\beta)^2} + \mathbb{E} \sum_{j\in[k]} \frac{1}{M_{ij}} \frac{(N_{ij}-n'M_{ij}\pi_i)^2}{(N_i+k\beta)^2} + \frac{o(1)}{n}$$

$$= -\mathbb{E}\frac{(N_i-n'\pi_i)^2}{(n'\pi_i+k\beta)^2} + \mathbb{E} \sum_{j\in[k]} \frac{1}{M_{ij}} \frac{(N_{ij}-n'\pi_i M_{ij})^2}{(n'\pi_i+k\beta)^2} + \frac{o(1)}{n}.$$

Finally, taking out $M(\delta)/6$, the last term can be bounded as

$$
\mathbb{E} \sum_{j \in [k]} \frac{N_{ij} + \beta}{N_i + k\beta} \left| \frac{(N_{ij} - M_{ij} N_i) + \beta(1 - kM_{ij})}{M_{ij}(N_i + k\beta)} \right|^3 \mathbb{1}_{A_i}
$$

$$
\leq 4 \sum_{j \in [k]} \frac{\mathbb{E} |N_{ij} - M_{ij} N_i|^3 + |\beta(1 - kM_{ij})|^3}{M_{ij}^3 (n'\pi_i(1 - \delta/2) + k\beta)^3} \mathbb{1}_{A_i}
$$

$$
\leq 4 \sum_{j \in [k]} \frac{4\mathbb{E} |N_{ij} - M_{ij} n'\pi_i|^3 + 4M_{ij}^3 \mathbb{E} |n'\pi_i - N_i|^3 + |\beta(1 - kM_{ij})|^3}{M_{ij}^3 (n'\pi_i(1 - \delta/2) + k\beta)^3} \mathbb{1}_{A_i}
$$

$$
= \frac{o(1)}{n},
$$

where we have used the ineuqality $(a + b)^3 \leq 4(|a|^3 + |b|^3)$ twice.

By the definition of $h(\cdot)$, we have

$$
h'(0) = -f'(0)
$$

and

$$
\frac{h''(0)}{2} = f'(0) + \frac{f''(0)}{2}.
$$

Hence, consolidating all the previous results,

$$
\mathbb{E}[D_f(M(i, \cdot), \hat{M}_{X^n}^{+\beta}(i, \cdot))]
$$

$$
= \frac{f''(0)}{2(n'\pi_i + k\beta)^2} \mathbb{E}\left( -(N_i - n'\pi_i)^2 + \sum_{j \in [k]} \frac{1}{M_{ij}} (N_{ij} - n'\pi_i M_{ij})^2 \right) + \frac{o(1)}{n}
$$

$$
= \frac{f''(0)}{2(n'\pi_i + k\beta)^2} \left( -\mathbb{E}N_i^2 + \sum_{j \in [k]} \frac{1}{M_{ij}} \mathbb{E}N_{ij}^2 \right) + \frac{o(1)}{n}.
$$

It remains to analyze $\mathbb{E}N_i^2$ and $\mathbb{E}N_{ij}^2$.

For $\mathbb{E}N_i^2$, we have

$$
\mathbb{E}N_i^2 = \mathbb{E}\left( \sum_{t<n} \mathbb{1}_{X_t = i} \right)^2
$$

$$
= \mathbb{E}\left( \sum_{t<n} \mathbb{1}_{X_t = i} \right) + 2\mathbb{E}\left( \sum_{t_1 < t_2 < n} \mathbb{1}_{X_{t_1} = i} \mathbb{1}_{X_{t_2} = i} \right)
$$

$$
= \sum_{t<n} \Pr(X_t = i) + 2 \sum_{t_1 < t_2 < n} \Pr(X_{t_1} = i) \Pr(X_{t_2} = i | X_{t_1} = i)
$$

$$
= n'\pi_i + \mathcal{O}(1) + 2 \sum_{t_1 < t_2 < n} \left( \pi_i \pm \mathcal{O}\left( \frac{1}{n^2} \right) \right) \Pr(X_{t_2} = i | X_{t_1} = i)
$$

$$
= n'\pi_i + \mathcal{O}(1) + 2\pi_i \sum_{t_1 < t_2 < n} \Pr(X_{t_2} = i | X_{t_1} = i)
$$

$$
= n'\pi_i + \mathcal{O}(1) + 2\pi_i \sum_{t_1 < t_2 < n} \sum_{j \in [k]} \Pr(X_{t_2} = i | X_{t_1+1} = j) \Pr(X_{t_1+1} = j | X_{t_1} = i)
$$

$$
= n'\pi_i + \mathcal{O}(1) + 2\pi_i \sum_{j \in [k]} \sum_{t_1 < t_2 < n} \Pr(X_{t_2} = i | X_{t_1+1} = j) M_{ij}.
$$

For $\mathbb{E}N_{ij}^2$, we have

$$\mathbb{E}N_{ij}^2 = \mathbb{E}\left(\sum_{t<n} \mathbb{1}_{X_t=i}\mathbb{1}_{X_{t+1}=j}\right)^2$$

$$= \mathbb{E}\left(\sum_{t<n} \mathbb{1}_{X_t=i}\mathbb{1}_{X_{t+1}=j}\right) + 2\mathbb{E}\left(\sum_{t_1<t_2<n} \mathbb{1}_{X_{t_1}=i}\mathbb{1}_{X_{t_1+1}=j}\mathbb{1}_{X_{t_2}=i}\mathbb{1}_{X_{t_2+1}=j}\right)$$

$$= M_{i,j}\sum_{t<n}\Pr(X_t=i) + 2\sum_{t_1<t_2<n}\Pr(X_{t_1}=i)M_{ij}\Pr(X_{t_2}=i|X_{t_1+1}=j)M_{ij}$$

$$= M_{i,j}n'\pi_i + \mathcal{O}(1) + 2\sum_{t_1<t_2<n}\left(\pi_i \pm \mathcal{O}\left(\frac{1}{n^2}\right)\right)\Pr(X_{t_2}=i|X_{t_1+1}=j)M_{ij}^2$$

$$= M_{i,j}n'\pi_i + \mathcal{O}(1) + 2\pi_i M_{ij}^2 \sum_{t_1<t_2<n}\Pr(X_{t_2}=i|X_{t_1+1}=j).$$

Thus, the desired quantity evaluates to

$$-\mathbb{E}N_i^2 + \sum_{j\in[k]}\frac{1}{M_{ij}}\mathbb{E}N_{ij}^2 = \sum_{j\in[k]}\left(n'\pi_i + \mathcal{O}(1) + 2\pi_i M_{ij}\sum_{t_1<t_2<n}\Pr(X_{t_2}=i|X_{t_1+1}=j)\right)$$

$$-\left(n'\pi_i + \mathcal{O}(1) + 2\pi_i\sum_{j\in[k]}\sum_{t_1<t_2<n}\Pr(X_{t_2}=i|X_{t_1+1}=j)M_{ij}\right)$$

$$\leq (k-1)n'\pi_i + \mathcal{O}(k).$$

The above inequality yields

$$\mathbb{E}[D_f(M(i,\cdot), \hat{M}_{X^n}^{+\beta}(i,\cdot))]$$

$$= \frac{f''(0)}{2(n'\pi_i+k\beta)^2}\mathbb{E}\left(-(N_i-n'\pi_i)^2 + \sum_{j\in[k]}\frac{1}{M_{ij}}(N_{ij}-n'\pi_i M_{ij})^2\right) + \frac{o(1)}{n}$$

$$\lesssim \frac{(k-1)f''(0)}{2n\pi_i}.$$

This completes our proof for ordinary $f$-divergences.

## 4.4  $L_2$-divergence upper bound

Finally, we consider the $L_2$-divergence. Again, we assume that the sample sequence $X^n \sim (M)$ and $\mu$ satisfies

$$D_{L_1}(\pi,\mu) < \frac{1}{n^2}.$$

Instead of using an add-constant estimator, we use the following add-$\sqrt{N_i}/k$ estimator:

$$\hat{M}_{X^n}^{+\sqrt{N_i}/k}(i,j) := \frac{N_{ij}+\sqrt{N_i}/k}{N_i+\sqrt{N_i}}, \ \forall i,j \in [k]\times[k].$$

Now, consider the expected loss for a particular state $i \in [k]$.

$$\mathbb{E}\sum_{j\in[k]}\left(M_{ij}-\frac{N_{ij}+\sqrt{N_i}/k}{N_i+\sqrt{N_i}}\right)^2 = \sum_{j\in[k]}\mathbb{E}\left(\frac{(M_{ij}N_i-N_{ij})+\sqrt{N_i}(M_{ij}-1/k)}{N_i+\sqrt{N_i}}\right)^2$$

$$= \sum_{j\in[k]}\mathbb{E}\left(\frac{M_{ij}N_i-N_{ij}}{N_i+\sqrt{N_i}}\right)^2 + \left(\frac{\sqrt{N_i}(M_{ij}-1/k)}{N_i+\sqrt{N_i}}\right)^2$$

$$+ 2\mathbb{E}\frac{(M_{ij}N_i-N_{ij})(\sqrt{N_i}(M_{ij}-1/k))}{\left(N_i+\sqrt{N_i}\right)^2}.$$

We first show that the last term is negligible. Noting that

$$\mathbb{E}\sum_{j\in[k]}\frac{(M_{ij}N_i-N_{ij})(\sqrt{N_i}(M_{ij}-1/k))}{\left(N_i+\sqrt{N_i}\right)^2}=\mathbb{E}\sum_{j\in[k]}\frac{(M_{ij}N_i-N_{ij})M_{ij}}{\sqrt{N_i}\left(\sqrt{N_i}+1\right)^2},$$

we can apply Taylor expansion to the function

$$f(x):=\frac{1}{\sqrt{x}(\sqrt{x}+1)^2}$$

at point $x=\mathbb{E}[N_i]$ and set $x=N_i$:

$$f(x)=f(\mathbb{E}[N_i])+f'(N_i')(N_i-\mathbb{E}[N_i]),$$

where $N_i'\in[\mathbb{E}[N_i],N_i]$. Hence,

$$\mathbb{E}\sum_{j\in[k]}\frac{(M_{ij}N_i-N_{ij})M_{ij}}{\left(N_i+\sqrt{N_i}\right)\left(\sqrt{N_i}+1\right)}$$

$$=\mathbb{E}\sum_{j\in[k]}\left(f(\mathbb{E}[N_i])+f'(N_i')(N_i-\mathbb{E}[N_i])\right)(M_{ij}N_i-N_{ij})M_{ij}$$

$$=\mathbb{E}\sum_{j\in[k]}\frac{(M_{ij}N_i-N_{ij})M_{ij}}{\sqrt{\mathbb{E}[N_i]}(\sqrt{\mathbb{E}[N_i]}+1)^2}+\frac{-3\sqrt{N_i'}-1}{2(\sqrt{N_i'}+1)^3(N_i')^{3/2}}(N_i-\mathbb{E}[N_i])(M_{ij}N_i-N_{ij})M_{ij}$$

$$\le\mathbb{E}\sum_{j\in[k]}\mathcal{O}\left(\frac{1}{n^{7/2}}\right)+\frac{-3\sqrt{N_i'}-1}{2(\sqrt{N_i'}+1)^3(N_i')^{3/2}}M_{ij}\sqrt{\mathbb{E}(N_i-\mathbb{E}[N_i])^2\mathbb{E}(M_{ij}N_i-N_{ij})^2}$$

$$=\Theta\left(\frac{1}{n^{3/2}}\right).$$

where the last step follows from Lemma 18. It remains to consider

$$\mathbb{E}\left(\frac{M_{ij}N_i-N_{ij}}{N_i+\sqrt{N_i}}\right)^2=\frac{\mathbb{E}(M_{ij}N_i-N_{ij})^2}{(n\pi_i+\sqrt{n\pi_i})^2}+\frac{o(1)}{n}.$$

According to the previous derivations, for $M_{ij}^2\mathbb{E}N_i^2$, we have

$$M_{ij}^2\mathbb{E}N_i^2=M_{ij}^2\sum_{t<n}\Pr(X_t=i)+2M_{ij}^2\sum_{t_1<t_2<n}\Pr(X_{t_1}=i)\Pr(X_{t_2}=i|X_{t_1}=i).$$

For $\mathbb{E}N_{ij}^2$, we have

$$\mathbb{E}N_{ij}^2=M_{ij}\sum_{t<n}\Pr(X_t=i)+2M_{ij}^2\sum_{t_1<t_2<n}\Pr(X_{t_1}=i)\Pr(X_{t_2}=i|X_{t_1+1}=j).$$

For $2M_{ij}\mathbb{E}N_{ij}N_i$, we have

$$2M_{ij}\mathbb{E}N_{ij}N_i=2M_{ij}\mathbb{E}\left(\sum_{t<n}\mathbb{1}_{X_t=i}\mathbb{1}_{X_{t+1}=j}\right)\left(\sum_{t<n}\mathbb{1}_{X_t=i}\right)$$

$$=2M_{ij}\mathbb{E}\left(\sum_{t<n}\mathbb{1}_{X_t=i}\mathbb{1}_{X_{t+1}=j}\right)+2M_{ij}\mathbb{E}\left(\sum_{t_1<t_2<n}\mathbb{1}_{X_{t_1}=i}\mathbb{1}_{X_{t_1+1}=j}\mathbb{1}_{X_{t_2}=i}\right)$$

$$+2M_{ij}\mathbb{E}\left(\sum_{t_2<t_1<n}\mathbb{1}_{X_{t_1}=i}\mathbb{1}_{X_{t_1+1}=j}\mathbb{1}_{X_{t_2}=i}\right)$$

$$=2M_{ij}^2\sum_{t<n}\Pr(X_t=i)+2M_{ij}^2\sum_{t_1<t_2<n}\Pr(X_{t_2}=i|X_{t_1+1}=j)\Pr(X_{t_1}=i)$$

$$+2M_{ij}^2\sum_{t_2<t_1<n}\Pr(X_{t_1}=i|X_{t_2}=i)\Pr(X_{t_2}=i).$$

Therefore,

$$\mathbb{E}(M_{ij}N_i - N_{ij})^2 = M_{ij}(1 - M_{ij})n\pi_i + \frac{o(1)}{n}.$$

Finally,

$$\mathbb{E}\sum_{j\in[k]}\left(\frac{\sqrt{N_i}(M_{ij} - 1/k)}{N_i + \sqrt{N_i}}\right)^2 = \frac{o(1)}{n} + \frac{-\frac{1}{k}\mathbb{E}[N_i] + \mathbb{E}[N_i]\sum_{j\in[k]}M_{ij}^2}{(n\pi_i + \sqrt{n\pi_i})^2}.$$

We have

$$\mathbb{E}\sum_{j\in[k]}\left(M_{ij} - \frac{N_{ij} + \sqrt{N_i}/k}{N_i + \sqrt{N_i}}\right)^2 = \left(1 - \frac{1}{k}\right)\frac{1}{n\pi_i} + \frac{o(1)}{n}.$$

This completes our proof for the $L_2$-divergence.