[Reviews · NeurIPS 2018]

Reviewer 1



# Review for "On Learning Markov Chains" This paper concerns two fundamental question about Markov chains. Consider observing $n$ steps of an unknown Markov chain on $k$ states, then doing one of the following: (1) predicting the distribution of the next state. (2) estimating the entire matrix of transition probabilities of the chain. How does the performance of minimax-optimal estimators for these tasks depend on $n$ and $k$? To make this question precise requires choosing a loss with which to measure the performance of the predictor. This paper makes substantial progress on question (1) when the loss is the KL divergence between the predicted distribution and true distrbution of the next state. It almost closes the book on question (2) for a variety of interesting losses, including KL divergence, chi^2 divergence, and Hellinger divergence. To give an idea of the flavor of results, one of their theorems is the following: the minimax-optimal loss for predicting the distribution of the n+1-st state of a k-state markov chain is between $C (k-1) \log \log n / n$ and $C' k^2 \log \log n /n$, for some constants $C$ and $C'$. The best previously-known result [Learning Markov Distributions: Does Estimation Trump Compression?, ISIT 2016] was that this risk is $\Theta_k(\log \log n / n)$; the dependence on $k$ was unknown. However, especially in machine learning applications the number of states of a Markov chain may be large; studying the dependence of the optimal estimation rates on $k$ is important. This paper is quite well written; I did not have time to verify the technical sections but the exposition in the first few pages is clear and convincing. The paper makes significant progress on core problems for the NIPS community. I strongly recommend that it be accepted.

Reviewer 2



Summary: The paper’s goal is to study the minimax rates for learning problems on Markovian data. The author(s) consider an interesting setting where the sequence of data observed follow a Markovian dependency pattern. They consider discrete state Markov chains with a state space [k] and study the minimax error rates for the following two tasks: Prediction: Given a trajectory X_1 -> X_2 … -> X_n from an unknown chain M, predict the probability distribution of the next state X_n+1, i.e., P(. | X_n). Measured by Prediction risk = E[ Loss(P(. | X_n) , P_hat(. | X_1…n)) ] where the expectation is over the trajectory X_1…X_n. The loss function the paper focuses on is KL-divergence and presents a conjecture for how the L_1 loss should scale with respect to k and n. Estimation: Given a trajectory X_1 -> X_2 … -> X_n from the unknown chain M, estimate the transition matrix M (call it M_hat). Goal is do well under the following loss: Estimation risk = max_{i \in [k]} E[ Loss( M( i , . ) , M_hat( i , . )) ] If the Markov chain is such that some states are observed infrequently, this affects the estimation problem but not so much the prediction problem as a trajectory ending in such a state will be rarely observed and we only want to do well on the average trajectory hence we can afford to do badly on such ‘rare’ states. Summary of results: The paper has upper and lower bounds for the minimax rates for the prediction task which are off by a factor of k. The authors conjecture that the right dependence is probably that of the upper bound. The paper gives tight results for the estimation problem for loss functions which are ordinary f-divergences and also has additional results for learning under l_2 loss. Overall the paper is well-written and presents the results well. It uses some interesting tools giving bounds on surprise probabilities and concentration inequalities for number of visits to a state in a Markov chain. Adding some intuition as to why the stated minimax rates are achieved would be nice. Comments: It would be nice to have a more comprehensive related work section (at least in the supplementary material). The asymptotic analysis of the minimax rates for the prediction problem seem to be ignoring an important term which decays as O_k(1/n) in the interest of the marginally dominating term O_k(loglog n/ n). Ignoring the 1/n decay term and quantifying the exact dependence on k with respect to the loglogn/n seems to not really be capturing the whole picture with respect to the dependence on k. For instance if the 1/n decaying term has a k^3 dependence for instance, then this term would dominate the sample complexity until n > 2^2^k which can be huge. Since the paper’s main contribution for this problem appears to be identifying the range of dependencies on k, it feels like identifying the dependence on k for the O_k(1/n) term is also important. The estimation problem as formulated by the authors seems to be somewhat harsh in the notion of the risk they use. A more natural risk function to study would seem to be \sum_{i \in [k]} \pi_i * E[ Loss( M( i , . ) , M_hat( i , . )) ] as it gives less importance to states we rarely visit and hence are hard to estimate accurate transitions from anyway. Any justification for why the chosen risk function is the natural one? Moreover, the harsh risk function used intuitively makes the lower bound somewhat straightforward. The lower bound construction essentially hides differences in one node of the chain and ensures that it is the least frequently visited node then the total sample complexity . Also the upper bound doesn’t seem very surprising given that once we wait a mixing time period (the paper state ignore the first sqrt(n) samples but treats n as increasing with the mixing time and stationary probabilities remain fixed, hence this can be interpreted as waiting for a mixing time steps?) concentration results on the number of visits to a state kick in and the problem seems to reduce to solving k IID distribution estimation problems? Experiments: For the prediction task, it feels the right experiment to perform would have been one which scales up k the number of states in the chain as the main contribution of the paper compared to previous work is identifying the dependence on k. Also for the chosen small values of n (between 250 and 1000) it is not clear if the theoretical results for the prediction task are really being verified since a) the theory is developed in an asymptotic manner and 1000 is too small a sample size to see a difference of loglog n factors, and b) since the tailing term is O_k(1/n) which yields a similar scaling down behavior to that of O_k(loglog n/n) for the chosen k value and range of n. A discussion of the proof intuitions would be nice to have in the main body. It was hard to get intuition from reading the proofs in the supplementary material. Also some intuition on what might be the rates for ordinary f-divergences beyond KL for the prediction tasks would be nice to have. If the problem becomes much harder with other f-divergences, some intuition as to why that is the case might be helpful. Overall, I feel that this paper explores an interesting direction but the results seem a bit incremental/incomplete (for the prediction task) and somewhat straightforward for the estimation task given the choice of the risk function (bypasses having to deal with the Markovian nature of the data)

Reviewer 3



This paper deals with the estimation of the transition matrix of a time homogeneous Markov chain on a finite space, together with prediction (i.e., estimation of the distribution of the next random state). For prediction, the authors prove a lower and an upper bound for the minimax risk. These bounds match in terms of the number of observations, but not in terms of the size of the state space, even though they conjecture that their upper bound is tight, based on the numerical experiments. For estimation, the authors prove matching upper and lower bounds in a minimax sense, for a class of divergences between probability measures. The results seem correct (although many proofs are very technical and I have not checked them) and I believe the problem is relevant. However, the major issue with this work is the writing style (shortcuts, omitting important details, typos or serious lack of precision), to my opinion. Here are a few examples. 1. The authors do not deal with any Markov chains, only time homogeneous ones. 2. Line 35+Line 39: Repeated. 3. Line 83: ordinary. 4. In many statements, most constants are very artificial. For instance, in Theorem 1, the factors 4e, 2, or writing (k-1) instead of k, are irrelevant, given the use of the signs \lesssim, unless the authors carefully redefine that symbol. To my opinion, this artificially makes the results look more complicated (even more in Line 113, with the use of the number pi!) 5. Line 97: Any reference for this statement? 6. Line 109: Unless I am wrong, this statement is wrong. Only the lower bound is stronger than that of Theorem 2. 7. The whole section 3 is uninformative, to my opinion. It only introduces some (quite technical) definitions and notation that are only used in the proofs, without giving any intuition. 8. Line 170 (also in the supplementary material): I do not see why this triangle inequality is correct. It may even be wrong, but I am not sure. However, in order to prove the upper bound, it suffices to construct one and only one estimator \hat P, that takes the first value if K_n is satisfied, the second one otherwise. 9. Line 172: "One can show", but how? Or any reference? 10. Line 177 in the definition of L_n, a \cup is missing. Overall, my opinion is that this paper should be accepted if a substantial revision is performed by the authors.